# Cooperation and the evolution of hunter-gatherer storytelling

Daniel Smith [1,2], Philip Schlaepfer[3], Katie Major[4], Mark Dyble [5,6], Abigail E. Page [1], James Thompson[1], Nikhil Chaudhary [1], Gul Deniz Salali [1], Ruth Mace [1,7], Leonora Astete[8], Marilyn Ngales[8], Lucio Vinicius[1] & Andrea Bamberg Migliano [1]

Storytelling is a human universal. From gathering around the camp-fire telling tales of ancestors to watching the latest television box-set, humans are inveterate producers and consumers of stories. Despite its ubiquity, little attention has been given to understanding the function and evolution of storytelling. Here we explore the impact of storytelling on hunter-gatherer cooperative behaviour and the individual-level fitness benefits to being a skilled storyteller. Stories told by the Agta, a Filipino hunter-gatherer population, convey messages relevant to coordinating behaviour in a foraging ecology, such as cooperation, sex equality and egalitarianism. These themes are present in narratives from other foraging societies. We also show that the presence of good storytellers is associated with increased cooperation. In return, skilled storytellers are preferred social partners and have greater reproductive success, providing a pathway by which group-beneficial behaviours, such as storytelling, can evolve via individual-level selection. We conclude that one of the adaptive functions of storytelling among hunter gatherers may be to organise cooperation.

[1] Department of Anthropology, University College London, London WC1H 0BW, UK. [2] Bristol Medical School: Population Health Sciences, University of Bristol, Bristol BS8 2BN, UK. [3] AgtaAid, Letchworth SG6 1RD, UK. [4] Bristol Zoological Society, Bristol BS8 3HA, UK. [5] Jesus College, University of Cambridge, Jesus Lane, Cambridge CB5 8BL, UK. [6] Department of Zoology, University of Cambridge, Downing Street, Cambridge CB2 3EJ, UK. [7] School of Life Sciences, Lanzhou University, Lanzhou 730000, China. [8] Lyceum of the Philippines University, Community Outreach and Service Learning, Manila 1002, Philippines. Lucio Vinicius and Andrea Bamberg Migliano contributed equally to this work. Correspondence and requests for materials should be addressed to D.S. (email: daniel.smith.11@ucl.ac.uk) or to A.B.M. (email: a.migliano@ucl.ac.uk)

Cooperation is a central problem in biology[1, 2]. This is especially true in humans given the range of extensive cooperation observed, including food sharing[3, 4], allocare[5, 6] and political coalitions[7]. Adaptive explanations for cooperation —broadly defined as a behaviour which evolved to benefit others[8] —often focus on the 'free-rider problem'; that is, explaining how a behaviour which decreases fitness (at least in the short-term) can be evolutionarily advantageous. Many solutions to this problem have been proposed, such as kin selection[9], reciprocal cooperation[10], costly signalling[11] and indirect reciprocity[12], among others. However, even in situations where cooperation would be the best strategy for all involved, cooperation may not occur due to 'problems of coordination'. Under these circumstances, cooperation is not hindered by the potential for free-riding, but rather by a lack of common knowledge over the behaviour of others[13, 14]. Meta-knowledge is therefore required to solve these problems of coordination. In other words, it is not enough to know how to act in a given situation; individuals need to know that others also know how to act. While language is undoubtedly essential as a medium of communication for coordination[15], here we propose that storytelling in particular may have played an essential role in the evolution of human cooperation by broadcasting social and cooperative norms to coordinate group behaviour (see also refs. [16, 17]).

Storytelling is a human universal[18] which occurs spontaneously in childhood[19], while cross-cultural phylogenetic analyses have shown that folk stories may be highly conserved[20]. The universal presence and antiquity of storytelling indicates that it may be an important human adaptation[21–24]. Hunter-gatherer societies have strong oral storytelling traditions dictating social behaviour regarding marriage, interactions with in-laws, food sharing, hunting norms and taboos[25–27]. These stories appear to coordinate group behaviour and facilitate cooperation by providing individuals with social information about the norms, rules and expectations in a given society[15, 28, 29]. It has recently been argued that religion with high-gods is a form of fictional story that helped in the expansion of large-scale human cooperation[30]. However, moralistic high-gods cannot be the original form of norm-enforcing fiction in human societies, as phylogenetic reconstructions suggest that they only emerged after increased political complexity associated with agricultural expansion[31]. Furthermore, hunter-gatherers display widespread cooperation (such as camp-wide food sharing, rituals for conflict resolution and long-term cooperation with unrelated individuals), and, despite being inveterate storytellers[25, 29], mostly lack the belief in moralistic high-gods[32]. Although others have proposed that storytelling was an important step in human evolution[16, 21–24], this hypothesis remains largely untested using real-world empirical data. For these reasons, we decided to analyse the content and functions of storytelling in a hunter-gatherer population (the Agta).

Here we show that: (i) Agta stories convey messages of cooperation, sex equality and social egalitarianism; (ii) stories from other hunter-gatherer societies also appear designed to coordinate social behaviour and promote cooperation; (iii) individuals from camps with a greater proportion of skilled storytellers are more cooperative; (iv) skilled storytellers are preferred social partners

**Table 1 Agta stories**

| Story | Plot | Promoted social norms | Mechanism |
|---|---|---|---|
| The sun and the moon | There is a dispute between the sun (male) and the moon (female) to illuminate the sky. After a fight, where the moon proves to be as strong as the sun, they agree in sharing the duty —one during the day and the other during the night. | Sex equality, cooperation between the sexes | Calculation and comparison of payoffs to cooperation vs. competition |
| The wild pig and the seacow | Wild pig and seacow were best friends and always raced each other for fun. But the seacow injured his legs and could not run anymore. The wild pig was unhappy and carried the seacow to the sea. They could race each other again, pig on land and seacow in the sea. | Friendship, cooperation | Advantageous inequality aversion |
| The monkey and the giant | The monkey and his other animal friends would like to camp close to the river. However, there was a giant there who would attack whoever went close to the river. They went anyway, and had to take turns to look after the camping site during the night. The giant came and said to the monkey he was going to eat them. Together they plot a defence plan against the giant: the monkey tricked the giant into a cave where they had hidden bee and ant nests. The giant died. The monkey was the leader of the plan. His friends congratulated him, but reminded him that even though he was the smartest animal in the forest, he was still vulnerable, as the monkey-eating eagle could take him. | Cooperation, social equality | Reverse dominance hierarchy |
| The winged ant | An ant who had wings lived together with other ants. One day she said to herself: 'I am not their friends because they don't have wings'. She went to bird and said, 'You must be my friend because you have wings.' Bird said, 'No you are an ant and I am a bird'. Then she went to the wasp, mosquito and butterfly and they all said the same. Then she went back to an ant and said, 'You must be my friend even though you don't have wings.' The Ant said, 'Yes you are an ant and I am an ant'. So all the ants welcomed her and said, 'Ant with wings, you are our queen'. | Social equality, group cohesion | Social acceptance, group identity |

The table shows four Agta stories, the main plot, promoted social norms and proposed mechanisms for norm compliance found in these stories

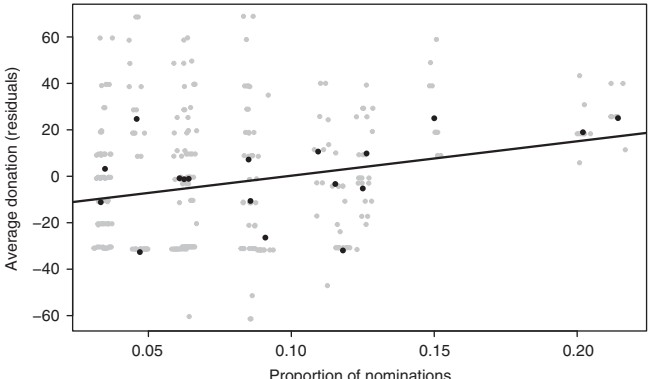

**Fig. 1** Association between cooperation and storytellers in camp. Results of the multi-level linear regression model, indicating an association between the average proportion of nominations for being a skilled storyteller in each camp and the amount given to others in the resource allocation game ($b$ = −215.6, 95% CI: [−47.8; −383.4], $p$ = 0.012). To facilitate interpretation, a higher value on the y-axis indicates a greater proportion of resources given to others (i.e. more cooperation). Residual values control for average camp relatedness, camp size and municipality ($n$ = 290, camps = 18). For coefficients of the full model see Supplementary Table 3. Black points: camp averages; grey points: individual data points

and more likely to be cooperated with and (v) skilled storytellers possess greater reproductive success. We conclude that story-telling may perform an adaptive function by organising coop-erative systems in hunter-gatherer societies. These results also provide a pathway by which group-beneficial behaviours, such as storytelling, can evolve via individual-level selection.

## Results

**Hunter-gatherer stories coordinate social behaviour**. We col-lected stories in a community of Agta in Palanan, the Philippines. We asked three elders to tell us stories they normally tell children and each other. The elders told us four stories over three nights (Table 1). All stories featured humanised natural entities such as animals or celestial bodies, but no Agta. All stories conveyed norms and principles regulating cooperation and social beha-viour, specifically sex equality ('The sun and the moon'), social egalitarianism and friendship ('The wild pig and the seacow'), group cooperation ('The monkey and the giant') and group identity and social acceptance ('The winged ant'). In these stories the ending reflects a reconciliation of individual interests and differences, while also exemplifying various mechanisms of social norm enforcement, such as emphasising the benefits to coop-eration over competition, examples of punishment for breaking norms, and reverse dominance hierarchies to prevent individual accumulation of power[16, 33]. These topics are recurrent in stories from other hunter-gatherer groups (Supplementary Table 1). To explore the content of hunter-gatherer stories in greater detail we collected 89 stories over seven different forager societies and coded them according to subject matter (see Methods for further details). Of these stories, around 70% were classified as pertaining to 'social behaviour' (i.e. prescribing social norms or coordinating behavioural expectations), more than any other category (Sup-plementary Table 2). Therefore, storytelling in general may pro-vide a mechanism to coordinate behaviour and expectations, transmit social information and promote cooperation in hunter-gatherer camps.

**Storytelling is associated with camp-level cooperation**. Con-firming this expectation, we found that camps with a greater

proportion of skilled storytellers were associated with increased levels of cooperation. We measured individual reputations for storytelling by asking people ($n$ = 297, over 18 camps in two municipalities) to name the best storytellers in camp (we imposed no lower or upper limit on the number of nominations; see 'Methods'). For each camp, the average proportion of nomina-tions received by each individual was used as a proxy for camp-level storytelling skill. We also asked 290 Agta adults from these camps to play an experimental resource allocation game in which subjects could either keep or share any desired fraction of a resource[34]. A multi-level linear regression model controlling for camp size, average camp relatedness and municipality showed that a greater proportion of individuals nominated as storytellers in a camp was significantly associated with increased coopera-tiveness ($b$ = −215.6, 95% CI: [−47.8; −383.4], $p$ = 0.012: Fig. 1, Supplementary Table 3). The regression coefficient implies that a 1% increase in nominations of good storytellers was associated with an increase in donations by 2.2 percentage points. This association is consistent with skilled storytellers spreading coop-erative norms and promoting cooperation in camps. However, this association may have alternative explanations and result from other social processes. For example, more cooperative camps may tell a greater number of stories, perhaps because they are more socially cohesive (although in Supplementary Table 4 we demonstrate that this result is unlikely to be an artefact of camp-mate familiarity, as these findings hold when controlling for the frequency of repeated interactions in a sub-set of camps). Therefore, if storytelling plays a functional role in promoting cooperation, we predict that skilled storytellers would be pre-ferred as social partners. In contrast, if storytelling is only a consequence or by-product of cooperation, preferred social partners are likely to be chosen on the basis of other character-istics, such as foraging skill or medicinal knowledge.

**Skilled storytellers are preferred social partners**. Since living in a more cooperative camp brings benefits at individual levels (even for non-cooperators), we tested whether people would prefer to live in camps with more skilled storytellers, where norms of cooperation are more likely to be spread. To assess storytelling reputation, for each camp the number of nominations received by each individual was converted into z-scores and transformed into a binary response variable ('skilled' vs. 'non-skilled' storyteller; see 'Methods'). We asked 291 Agta across the 18 camps to choose who they would most like to live with (with a maximum of five nominations), obtaining 857 nominations out of a possible 6534 dyads (all of which were within-camp nominations). We ran a logistic generalised estimation equation (GEE) regression to predict the probability of being picked as a future camp mate from the measure of individual storytelling reputation. We found that skilled storytellers were nearly twice as likely to be nominated as less skilled individuals (Table 2, model 1). This pattern holds after controlling for kinship, reciprocal nominations, distance, as well as age and sex variables (Table 2, model 2). In addition to storytelling, other reputational measures were assessed, including skill in hunting, fishing, tuber gathering, medicinal knowledge and camp influence (Methods). Including these factors in the model (Table 2, model 3) indicated that storytelling was the most important reputational attribute, with skilled storytellers again having roughly double the odds of being nominated relative to non-skilled storytellers (OR = 1.95), an effect much larger than that of possessing a good fishing reputation, the second-best reputational predictor (OR = 1.5). Removal of all non-significant variables does not alter these findings (Supplementary Table 5). The effect size of storytelling ability was approximately the same magnitude as selecting primary kin (PK) and reciprocal partners.

**Table 2 Storytelling ability and camp-mate decisions**

| Variable | Model 1 | | Model 2 | | Model 3 | |
|---|---|---|---|---|---|---|
| | Log-odds estimate | Odds ratio | Log-odds estimate | Odds ratio | Log-odds estimate | Odds ratio |
| Storytelling reputation | 0.56 [0.4; 0.72]*** | 1.75 | 0.78 [0.58; 0.98]*** | 2.17 | 0.67 [0.47; 0.87]*** | 1.95 |
| Primary kin (ref. non-kin) | — | — | 0.74 [0.35; 1.13]*** | 2.09 | 0.7 [0.31; 1.09]*** | 2.02 |
| Distant kin (ref. non-kin) | — | — | 0.59 [0.3; 0.88]*** | 1.81 | 0.57 [0.28; 0.86]*** | 1.77 |
| Spouse's primary kin/primary kin's spouse (ref. non-kin) | — | — | 0.59 [0.24; 0.94]*** | 1.81 | 0.57 [0.22; 0.92]** | 1.77 |
| Spouse's distant kin/other affines (ref. non-kin) | — | — | 0.25 [0; 0.5]* | 1.29 | 0.24 [−0.01; 0.49]˙ | 1.27 |
| Spouse (ref. non-kin) | — | — | −0.25 [−0.98; 0.48] | 0.78 | −0.28 [−1.02; 0.46] | 0.76 |
| Reciprocity | — | — | 0.66 [0.46; 0.86]*** | 1.93 | 0.67 [0.47; 0.87]*** | 1.95 |
| Fishing reputation | — | — | — | — | 0.4 [0.13; 0.67]** | 1.5 |
| Hunting reputation | — | — | — | — | 0.24 [−0.03; 0.51]˙ | 1.27 |
| Tuber gathering reputation | — | — | — | — | 0.27 [0; 0.54]˙ | 1.31 |
| Medicinal knowledge reputation | — | — | — | — | 0.12 [−0.12; 0.36] | 1.12 |
| Camp influence reputation | — | — | — | — | 0.23 [−0.02; 0.48]˙ | 1.26 |
| Intercept | −0.88 [−0.68; −1.08]*** | | 0.92 [0.35; 1.49]** | | 0.97 [0.38; 1.56]** | |
| Distance, age, and sex controls | No | | Yes | | Yes | |

Models assessing the likelihood of skilled storytellers being selected in a 'camp-mate' network, using a logistic GEE regression ($n = 291$, dyads = 6534). All models contain camp size as a control variable (not displayed). 95% confidence intervals are displayed in brackets. ˙$P < 0.1$, *$P < 0.05$, **$P < 0.01$, ***$P < 0.001$

These results demonstrate that the Agta prefer to live in camps with skilled storytellers, who are even more valued than good foragers, which may reflect the importance of storytellers in promoting cooperation and bringing gains to all individuals in a camp (although storytellers may also be favoured for disseminating other fitness-relevant information as well, such as foraging, survival and geography[16, 26, 35, 36]). Although these results point to a group-level advantage of storytelling, they do not indicate what would be the individual benefit for storytellers (although see ref. [24]). In other words, they do not explain why individuals would invest in acquiring a costly skill with no apparent individual fitness benefits compared to other skills such as hunting, gathering or fishing.

**Skilled storytellers have higher reproductive success.** Storytelling is a costly behaviour requiring an input of time and energy into practice, performance and cognitive processing[37, 38]. Indeed, several ethnographic sources highlight the theatrical and active nature often associated with storytelling performances[27, 29]. In addition, the group-level benefits of storytelling are susceptible to free-riders, who could reap the benefits of storytelling without paying the costs[1]. Thus, all else being equal non-storytellers should have higher fitness, unless storytelling brings direct fitness benefits to skilled storytellers. We therefore investigated reproductive success as a function of storytelling skill among the Agta. We ran a mixed-effects linear regression of number of living offspring on storytelling ability (controlling for age, sex and camp: Supplementary Table 6). The results show that skilled storytellers had an additional 0.53 living offspring compared to non-skilled storytellers ($b = 0.53$, 95% CI: [0.10; 0.96], $n = 324$, $p = 0.016$), indicating that storytelling skill is associated with increased fitness (Fig. 2). Both camp-mate nomination and fitness results are robust to manipulations, such as using continuous, rather than binary, assessments of storytelling ability, and quantifying storytelling ability separately for each sex (due to the potential for female Agta to be over-represented as skilled storytellers: see 'Methods' & Supplementary Tables 6–9).

It is possible that by performing an important social function skilled storytellers receive increased social support from others[39], which has been associated with increased fitness among numerous primate species[40], consistent with the fact that they are preferred social partners. Supporting this interpretation, we also demonstrate that skilled storytellers are more likely to be recipients of resource transfers in the experimental game (Supplementary Tables 10 and 11). This suggests that storytellers may be 'rewarded' for their public good by other camp mates who benefit from the increased cooperation which storytellers may promote, in what may be mutually beneficial trade-like relationships (although the individual-level benefits to those who cooperate with storytellers remain in need of further empirical study). People might also enjoy listening to stories for other reasons, such as a form of 'mental simulation' to learn about their social and physical environment[24, 41–47], and be paying for the service (this effect could be independent from the function of storytelling in promoting cooperation).

**Discussion**
We conclude that storytelling may perform an important adaptive function in hunter-gatherer societies by organising cooperative systems, serving the function of 'broadcasting' cooperative norms. Storytelling among the Agta and other hunter-gatherers conveys strong messages of cooperation, sex and social equality, and inequality aversion, and are widely told in camps to adults and children[25]. These stories appear to promote cooperation within a camp, as the proportion of skilled storytellers is positively associated with measures of camp-level cooperation. Furthermore, people show a strong preference to live with good storytellers, even more so than with good foragers, despite the fact that Agta society is characterised by extensive food sharing[4]. By introducing individuals to situations beyond their everyday experience, narratives may also increase empathy and perspective taking towards others, including strangers[48–50] (although see ref. [51]), potentially facilitating camp-level coordination and cooperation. The value of good storytellers is reflected in the fact that they also have increased reproductive success and receive more resources than less-skilled storytellers. We therefore provide a pathway by which storytelling, a group-beneficial behaviour, can evolve via individual-level selection (however, further research is necessary to understand the individual-level benefits of camp-mates cooperating with storytellers). Although narratives are known to serve other adaptive functions, such as disseminating information on survival, foraging and the environment[16, 26, 35, 36],

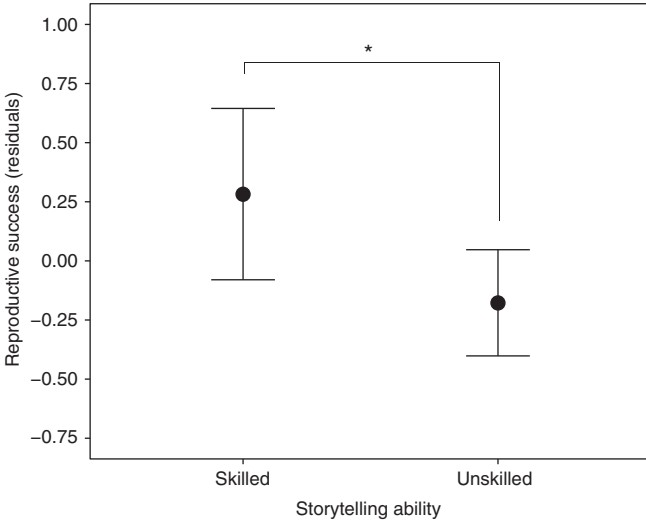

**Fig. 2** Reproductive success and storytelling ability. Results of the mixed-effects linear regression model, indicating that reproductive success, based on number of living offspring, was greater in skilled storytellers ($n = 125$) relative to less-skilled storytellers ($n = 199$; $b = 0.53$, 95% CI: [0.10; 0.96], $p = 0.016$). Residuals control for age, age-squared, sex and camp. Error bars represent 95% confidence intervals

or simply to entertain and hold the audience's attention[23], we provide evidence that storytelling may have also been an important factor in facilitating widespread human cooperation. It is therefore possible that cooperation and storytelling co-evolved via a process of mutual reinforcement.

The evidence provided here is consistent with the theory that storytelling acts as a mechanism to coordinate group behaviour and promote cooperation. However, these findings are largely correlational and further studies are required to conclusively demonstrate that storytelling performs a causal role in facilitating cooperative behaviour. One potential option would be to conduct longitudinal work to explore if patterns of camp-level cooperation vary with changes in camp composition, explicitly regarding the addition or loss of skilled storytellers. Logistically, however, this approach is very demanding, therefore a simpler approach to establish causality may be to use 'priming' experiments, such that those primed with a cooperative story ought to cooperate more in a subsequent task. Similar studies have previously been conducted (although not explicitly regarding storytelling), suggesting that individuals in experimental games are more cooperative if they can communicate and therefore coordinate their behaviour[15, 52], consistent with the mechanism proposed here. Additionally, although we demonstrate an association between the presence of skilled storytellers in camp and levels of cooperation, future studies may benefit from utilising more direct indices of storytelling, such as the amount of storytelling in a camp, to assess this putative link in greater detail.

Humans have evolved the cognitive ability to create and believe in stories. We propose that those features evolved in hunter-gatherer societies as precursors to more elaborate forms of narrative fiction, such as moralising high-gods. In hunter-gatherer societies, which are highly egalitarian[53, 54] and cooperative[4, 55, 56], but where organised religion and moralising high-gods are commonly absent[32], storytelling appears to promote cooperation, spread cooperative norms[57] and represent punishment of norm-breakers. In sum, we argue that storytelling may perform an adaptive function by organising cooperation in hunter-gatherers, preceding the emergence of more complex fictions such as all-knowing and punitive high-gods associated with the expansion of large-scale cooperation in human societies only after the origin of

farming[30]. From simple storytelling to complex religion, and later formal institutions such as nation states, the evolution of storytelling may have been pivotal in organising and promoting human cooperation.

## Methods

**Ethnographic background.** The Agta are an indigenous Filipino population, believed to be descendants of the first colonisers of the Philippines over 35,000 years ago[58], and are distinguishable from their non-Agta neighbours by their short 'pygmy' physique, dark skin, tight curly hair and predominantly foraging mode of subsistence. The Agta in the study population are from the Northern Sierra Madre Natural Park in Isabela Province, north-east Luzon, a remote area of protected forest land accessible only by plane, boat, or a three-day hike. Specifically, the study focuses on two sub-populations, the Palanan Agta, who number around 1,000 individuals, and the Maconacon Agta, who number around 250 individuals. These sub-populations live ~50 km apart and are largely separate, with few genealogical links between the two. Both sub-populations live on or near river-banks or coastal areas, and predominantly engage in foraging activities, particularly fishing, but also hunting, collecting honey, and gathering wild plants, which they either consume or trade for rice with the local agricultural non-Agta population. Depending on availability, some Agta also participate in wage labour (often clearing land for non-Agta farmers) or assist in rice harvesting, where they often receive a share of the harvest. Camp sizes vary between solitary dwellings (7 individuals) and large camps of up to 26 houses (156 individuals), with an average of 7 houses (49 individuals). Although many Agta nominally classify themselves as Christian (of various denominations), few Agta regularly attend church or possess much knowledge about Christianity. A few camps are more integrated with organised religion, particularly due to Born Again Christian Missionaries, where traditional beliefs in 'bad spirits' (*anitos*) are combined with Christianity. In traditional Agta cosmology there are no moralising or punishing high-gods. Initial demographic fieldwork was conducted between April and June 2013, with further fieldwork conducted between February and October 2014.

**Storytelling definition.** 'Storytelling' is defined loosely here to encompass a spectrum of narrative forms, from 'ritualised' storytelling, often in larger groups and accompanied by cosmological or religious content, to less-structured story-telling as a part of everyday conversation among smaller groups. Cutting across these contextual differences, a story can be broadly defined as 'an account of a sequence of events in the order in which they occurred to make a point'[59]. A story can also be defined by its components, and several lines of evidence have converged from literary theory and cognitive psychology which suggest that narratives consist of: character, setting, events, causal connections and resolution (for a review see ref. [24]). These are inclusive definitions of 'storytelling', which encompass both ritualised or fictional stories, as well as many aspects of everyday conversation which also contain non-fictional narratives, such as jokes, anecdotes or details of previous experiences. Although much research focuses on fictional stories[29], the importance of non-fictional narratives should not be overlooked. For instance, in her work with the Ju/'hoansi, Wiessner[25] provides details of several stories, all of which are non-fictional, which broadcast social norms concerning issues such as sex, marriage, sharing obligations and norm-breakers (see also Supplementary Table 1 for other examples of non-fictional stories, such as by the Batak).

**Assessing storytelling reputation.** To assess storytelling reputation, 297 Agta over 18 camps (mean age = 37, range = 16–70, males = 142) were asked to name the best storytellers in camp (there was no limit on the number of names individuals could select). From this, storytelling ability was calculated for 324 Agta. To permit comparisons between camps of different sizes the number of nominations for each individual were converted into z-scores for each camp separately (calculated by subtracting the camp mean from the individual score, then dividing this by the camp standard deviation). Due to a heavily-skewed distribution (lots of poor, but few good, storytellers; Supplementary Fig. 1), these were converted into a binary variable, with skilled storytellers being those above the mean for their camp (i.e. a positive z-score). From this, 199 Agta were classified as 'unskilled' storytellers (males = 109), while 125 were categorised as 'skilled' (males = 51). The average relatedness of individuals to camp-mates, relative to the overall camp relatedness, was no different between skilled and unskilled storytellers (mean relatedness relative to overall camp relatedness for skilled storytellers = 0.001, SE = 0.005, mean relatedness relative to overall camp relatedness for unskilled storytellers = −0.001, SE = 0.004; $t = −0.229$, $n = 324$, $p = 0.819$), suggesting that nominations for story-tellers were unaffected by relatedness.

One potential issue with this methodology is that it focuses on within-camp comparisons of storytelling ability, such that storytelling skill may be relative to each camp, rather than absolute for the population, which may hinder between-camp comparisons. However, this is unlikely to have occurred as the average proportion of nominations for each individual per camp ranged from 0.03 to 0.21 (see 'Resource Allocation Game' section below), indicative of substantial between-camp variation in the number of nominations for storytelling ability. Additionally, the proportion of individuals who nominated no-one varied considerably between camps (from 0% to 62.5%). A $\chi^2$ test of independence reported that the frequency

of individuals nominating no-one varied significantly between camps ($\chi^2(17$, $n = 297) = 82.28$, $p < 0.001$). Together, these suggest that storyteller nominations were not simply relative to camp-mates, but rather reflect absolute differences in skill.

In a logistic regression containing age and sex, older individuals possessed a greater storytelling reputation ($b = 0.04$, 95% CI: [0.02; 0.06], $n = 324$, $p < 0.001$). An interaction was also found between age and sex (Supplementary Fig. 2), with storytelling ability increasing more with age for men than for woman ($b = 0.04$, 95% CI: [0.01; 0.08], $n = 324$, $p = 0.033$). Women were also more likely to be skilled storytellers than men ($b = -0.76$ (ref. female), 95% CI: [−0.27; −1.25], $n = 324$, $p = 0.002$). While both sexes selected an equivalent proportion of same-sex individuals as skilled storytellers (female mean = 0.73, male mean = 0.69, Mann Whitney $U = 6,162$, $n = 297$, $p = 0.38$), females nominated a greater number of individuals overall (female mean = 1.97, male mean = 1.47, Mann Whitney $U = 8,760$, $n = 297$, $p = 0.004$), which may explain why women were over-represented as skilled storytellers relative to men. If $z$-scores for storytellers are calculated separately for each sex to control for the bias of females nominating other females, we find that the effect of sex on storytelling skill is largely attenuated, but still approaches significance in the direction of more skilled female storytellers ($b = -0.42$, 95% CI: [0.05; −0.89], $n = 324$, $p = 0.083$). The effect of age is still strongly significant ($b = 0.04$, 95% CI: [0.02; 0.06], $n = 324$, $p < 0.001$), while a significant interaction between age and sex is again reported if this is included in the model ($b = 0.04$, 95% CI: [0.00; 0.08], $n = 324$, $p = 0.04$).

In addition to assessing storytelling reputation, these 297 individuals were also asked to nominate the best hunters, fishers, tuber-gatherers, and those with the most medicinal knowledge. Camp influence was also assessed, but in a slightly different way, by asking 'if there is a discussion in camp, whose opinions are listened to the most? Who is *malakas* (strong)?' Individual $z$-scores were again constructed for individuals in each camp in these domains. Although there were no restrictions against naming females as hunters or fishers, or males as tuber-gatherers, these were very rare, so all instances were removed prior to analysis and $z$-scores for these domains were only constructed for the relevant sex (hunting and fishing for males, tuber gathering for females). To allow all prestige measures across all individuals to be compared in the same model, all women were assigned a '0' for hunting and fishing, while all men were given a '0' for tuber gathering. Although including sex as a co-variate in analyses should control for these sex-specific foraging domains, to remove this potential confound we constructed an additional 'overall foraging skill' variable combining both male and female foraging activities (using average number of hunting and fishing nominations for male foraging skill, tuber-gathering nominations for female foraging skill, then merging the two). Using this new variable, we find no qualitative difference in our main findings regarding who individuals chose as camp-mates or as recipients of cooperation (Supplementary Table 12), with storytellers still selected more frequently than skilled foragers.

Although there is some correlation between different reputational domains (Supplementary Table 13), these are generally weak ($r < 0.3$). Collinearity diagnostics cannot be conducted for GEE analyses, but a similar approach employing multiple regression using aggregate popularity for each individual ($z$-scores based on the number of nominations for each individual, calculating each camp separately) indicated that collinearity between these reputational domains is weak (all 'variance inflation factors' (VIFs) <1.5; a VIF greater than '3' is indicative of severe collinearity), and is therefore unlikely to bias these results.

Although it is difficult to validate all our reputational measures against actual behaviour, it was possible to assess a small sample of fishers from one particularly well-studied camp for which enough foraging trips were recorded to permit comparisons between individuals. In this camp, perceived skill in fishing was significantly correlated with both fishing returns per hour ($r = 0.606$, $n = 16$, $p = 0.013$) and total calories obtained from fishing ($r = 0.802$, $n = 16$, $p < 0.001$), indicating that these nominations likely reflect a combination of both effort and skill, and are therefore valid and can be used as a proxy for skill level. Further verifying this methodology, comparable protocols on Hadza hunting skill indicated a similar profile, with those perceived as possessing greater hunting skill having greater overall return rates and returns per hour[60]. Many of the Agta rated by others as skilled storytellers were also those who were the most engaging and knowledgeable during our fieldwork. These reputational measures therefore appear to reflect real-world dynamics among the Agta.

**Resource allocation game**. A simple resource allocation game was played with 290 Agta (mean age = 37, range = 16–70, males = 140) across these 18 camps to assess levels of cooperation. Only camps with eight and more adult members present were included in the games for the statistical analysis to possess an adequate sample size. Due to the majority of Agta not knowing their exact ages, adults were defined as either married or divorced individuals, or those believed to be over the age of ~16. Approximately ten days were spent at each Agta camp. Games were only played on the last few days in order to maximise familiarity with the researchers and facilitate trust, but also to minimise the potential for collusion between camp-mates. We do not believe that this occurred, as there were no sudden shifts in game behaviour over time. Prior to playing the game in each camp, photographs of all players were taken and Polaroids printed.

In private, participants were shown their own picture, along with all other camp-mates (up to a maximum of 10). Thus, in camps with 12 or more members,

10 randomly selected camp-mates (in addition to ego) were chosen. The decision to randomly-select 10 individuals from larger camps, rather than include all individuals, was chosen for practical and comparative reasons. Firstly, including all camp-mates from larger camps would have been logistically unfeasible as the amount of rice needed would increase exponentially with camp size. Secondly, although it would have been possible to limit the amount of rice by including all individuals in a large camp and only using ten tokens, this would make comparisons between larger and smaller camps difficult as otherwise the ratio of equal tokens to potential recipients would be violated in larger camps.

A number of tokens equal to the number of camp-mate pictures were then given to the participant, each of which represented one-eighth of a kilo of rice (125 g; approximately a meal for one individual). Participants were then asked, for each token, whether they would like to keep the rice for themselves, or give it to a camp-mate, and if so, to whom. After each iteration, tokens were placed on the respective individual. This was repeated until there were no tokens remaining. Only the participant, experimenter, and translator were aware of an individual's decisions. Individuals were briefed on the games in their local language, and assured that all decisions would remain secret from other camp-mates. They were told that there were no correct answers, and that they, and whoever they gave rice to, would be given it before the researchers left camp. After finishing the game, participants were thanked and asked politely not to tell anyone else about how they played. In total, the procedure took about 10–15 min per participant. Prior to leaving camp, the amount of rice earned by each participant was given to them (the amount they received from others and the amount they kept for themselves), along with remuneration for their time and assistance in other aspects of the project conducted simultaneously.

This non-anonymous game structure was used in order for both levels of cooperative behaviour and patterns of cooperation (i.e. who individuals share with, such as kin, reciprocal partners or storytellers) to be ascertained. The game is similar to the 'Gift Game' conducted in several populations[56, 61–63] where participants are given resources (e.g. sticks of honey) and have to decide who to give it to. Although the game used here is structurally alike, it possesses the added rule that participants could either keep a share for themselves if they wished, or give it to a camp-mate of their choosing. Although the Gift Game allows the choice of giving to multiple individuals, it does not measure levels of cooperation as there is no option for keeping gifts for one's self, and is therefore not a social dilemma[64] as there is no conflict between individual and group interests. On the other hand, although traditional economic games, such as the Ultimatum Game, Dictator Game, and Public Goods Game[65], are social dilemmas, they include only anonymous partners, and therefore ignore the role that differences in relationship have on cooperation[66] and cannot be used to explore who individuals preferentially share resources with.

After preliminary trials with different resources, it was decided that rice would be used as the game resource as it is highly sought-after by the Agta and therefore carries enough value to cause a dilemma when deciding whether to share or not. Initial trials with other goods, such as honey sticks, were perceived to have little value (and were freely distributed to children). After discussing with the Agta which resources were most valued, rice was the unanimous choice. The Agta do not grow their own rice (although they may harvest it for neighbouring agricultural populations), and although it is a non-foraged commodity introduced by non-Agta agricultural populations it is one of the Agta's primary sources of calories (when available) and is highly valued. The vast majority of meals are consumed with rice, and in some cases consist solely of rice.

The percentage of tokens kept for self was used as the dependent variable in this analysis, with a higher percentage meaning more rice kept for self and less given to others. This ranged from 0% to 100%, with a mean of 62.6%. Multi-level models were used to control for the non-independence of data points within camps[67]. The independent variable of interest was the average proportion of skilled storyteller nominations per individual for each camp, which ranged from 0.03 to 0.21, with a mean of 0.1. Proportion of nominations, rather than number of nominations, was used in this analysis to control for differences in camp size. For instance, in a camp of 10 individuals, if on average each camp-mate receives two nominations, the average proportion of nominations would be 0.2 per person. In contrast, for a camp of 20 individuals, an average of two nominations per individual only corresponds to an average proportion of nominations of 0.1 per person. Individuals from the first camp were therefore proportionally more likely to be nominated than individuals from the second group, suggesting that the first camp contains better storytellers. As group size[68] and relatedness[9] may influence cooperation, these were both included as fixed effects to control for these potential confounds. Municipality was also included as a dummy fixed effect to control for differences in cooperation between Palanan and Maconacon. An additional analysis was conducted with a sub-set of 11 camps from Palanan for which data on the frequency of repeated interactions were available to explore whether storytelling skill was confounded with camp-mate familiarity (Supplementary Table 4).

**Camp-mate and resource distribution network analyses**. To assess social ties, 291 Agta (mean age = 37.3 range = 16–70, males = 138) were asked to name the five individuals they would most like to live with (similar to the 'camp-mate network' conducted with the Hadza[61]). This was conducted in a separate interview to the reputational questions in an attempt to forestall cross-over effects. Once nominations for non-Agta, non-camp-mates, and other camp-mates who did not take

part in the camp-mate network were removed, a total of 857 nominations remained from a possible 6534 dyads. Logistic GEE regression methods were used to control for multiple nominations by the same individual[69].

For the response variable, a matrix was constructed containing a '1' if ego selected alter to live with or a '0' if not. Between-camp dyads were coded as missing. The main independent variable of interest was storytelling reputation, as defined above, with skilled storytellers coded as '1' and unskilled storytellers as '0'. Other predictor variables included: kinship, reciprocity (if alter chose to live with ego), distance, as well as age (of ego, alter, and age gap between ego and alter) and sex (of ego, alter, and whether ego and alter were of the same or different sex). Kin relationships were defined as: PK, with a relatedness coefficient of $r = 0.5$ to ego; distant kin (DK), with a relatedness coefficient between $r = 0.25$ to $r = 0.03125$ (second cousins) to ego; spouse; spouse's primary kin/primary kin's spouse (SPK/PKS); spouse's distant kin/other affines (SDK/OA), which includes DK of spouse or other affinal relationships up to five steps away from ego (e.g. spouse's brother's wife's mother (four steps away)); and non-relatives (NR), which includes everyone else without a kinship link to ego (for further details see ref. [53]). As these are categorical variables, each of these kinship categories were compared against the probability of selecting to live with non-kin. The matrix for reciprocity was the transpose of the response variable (i.e. whether alter chose to live with ego). Distance was coded from one to four, reflecting increasing distance between ego and alter, with categories of; living in the same house as ego (1), living in the house next to ego (2), having a house between ego's and alter's (3) and living further away (4). The relationship between nominating an individual in the camp-mate network and other reputational domains (hunting, fishing, etc.) were also assessed. Camp size was included as a control in all models to control for larger camps possessing a greater number of potential recipients to nominate.

The resource allocation network employed an identical logistic GEE regression approach using the same methodology and variables as described above, but here using nominations of who individuals distributed resources to in the resource allocation game ($n = 290$, dyads = 1312). In this analysis the proportion of resources kept for self was used as a control variable to ensure that patterns of resource distributions were not confounded with overall levels of cooperation.

**Reproductive success analysis**. Number of living offspring was used as a proxy for reproductive success. This was ascertained by obtaining reproductive histories of all individuals during genealogical interviews. Both age and age-squared were included in the model to control for differences in age-specific fertility. As husbands are generally older than their wives among the Agta, this may result in lower estimates of age-specific reproductive success for men, so sex was also included as a control. Fertility, mortality and reproductive success among the Agta also vary depending on the camp[70], so multi-level modelling was used to control for camp-level variation in reproductive success. When included, the interaction between storytelling skill and sex is non-significant ($b = 0.61$, CI: [−0.25; 1.47], $n = 324$, $p = 0.17$), suggesting that reproductive benefits of being a skilled storyteller may accrue to both sexes.

**Robustness checks**. In this section we demonstrate that these results of storytellers possessing an increased number of camp-mate nominations as well as higher fitness are not a statistical artefact, as we replicate results now using storytelling as a continuous variable (rather than binary), and controlling for female-biased nominations of storytellers by quantifying storytelling skill separately for each sex (both as a binary and continuous variable; see 'Assessing storytelling reputation' section of Methods above). The results for fitness outcomes (Supplementary Table 6) and camp-mate nominations (Supplementary Tables 7–9) are qualitatively identical to those presented in the main text, with skilled storytellers both possessing greater reproductive success and more likely to be nominated in a camp-mate network. For camp-mate models using continuous, rather than binary, measures of reputations, z-scores for sex-specific domains (hunting, fishing and tuber gathering) were constructed using data from both sexes.

**Stories from other hunter-gatherer societies**. We conducted a literature search in order to compare the content of Agta stories against those from other Southeast Asian hunter-gatherer societies. This included other hunter-gatherer groups such as the Andamanese, the Batek (from peninsula Malaysia), the Maniq (from Thailand), and other populations from the Philippines (the Batak, Aeta and neighbouring Agta groups; Supplementary Table 1). It is difficult to obtain an unbiased sample of stories told by hunter-gatherers as different ethnographers tend to focus on different story contents (when investigated at all), but we specifically selected stories that concern norms of social behaviour, such as cooperation, relationships between the sexes and social hierarchies. Stories by the Ju/'hoansi from southern Africa[25, 29] and Central African pygmies[27] were also sought out and included in our content analysis (Supplementary Table 2), although they were not included in Supplementary Table 1. Among the Ju/'hoansi many stories concern marriage behaviour, in-law relationships, kinship networks and sharing norms[25], as well as sex equality and the different (but complementary) roles of each sex[29, 71]. Similar trends are found among central African pygmy groups, such as the BaYaka[27], who have legends and rituals which reinforce norms of sexual egalitarianism and social behaviour, as well as interactions with non-BaYaka.

These stories told by hunter-gatherers appear to differ dramatically from those of non-foraging populations. For instance, among the Bantu in southern Africa most stories concern maintaining the status quo to preserve the leader's authority[71], which is quite the opposite of hunter-gatherer stories which tend to extol the virtues of egalitarianism and equality. Further supporting these differences, in hunter-gatherers facing increased settlement and agriculture, rituals have shifted towards practices facilitating hierarchy and sexual inequality[72].

Not all stories obtained from these populations concerned social behaviour. Many stories not described here concern cosmological content, with seemingly little social relevance. Examples include Batek origins of the cosmos, Maniq origins of night and day (originating from a snake continuously eating and regurgitating its tail), and Andamanese origins of pigs (see references in Supplementary Table 1). Their existence does not diminish the importance of stories in facilitating cooperation, but merely highlights that stories can perform other functions, such as disseminating fitness-relevant information[16, 26, 35] or just to hold the audience's attention[23]. It is also possible that these stories, regardless of content, also play a functional role by acting as 'ethnic markers' which identify group membership, again coordinating behaviour and promoting cooperation[73, 74], but this is a separate (and non-mutually exclusive) argument to that of the current paper.

In addition to the stories concerning social behaviour (Supplementary Table 1), we also performed a content analysis of all stories collected ($n = 89$) from these Southeast Asian hunter-gatherer populations in addition to two African forager groups (Ju/'hoansi and BaYaka; Supplementary Table 2). Each story was assessed for different types of content: social (content which prescribed and coordinated behaviour during interactions with others, such as cooperation, sex equality, norm-breaking, sex roles, punishment and interactions with out-groups); cosmological (content concerning the origins of the earth/universe); natural phenomena (content concerning navigation, animal origins/behaviour, fire and natural disasters/weather); and resource use (content about foraging and resource extraction). Each of these broad themes are common in forager folklore[16, 25, 29, 35, 75]. Many stories were classified along more than one criterion, such as many creation stories combining cosmological and social content (for instance, see Supplementary Table 1). Some sources presented several versions of the same basic story; to prevent unnecessary duplication, in these cases only the longest or most elaborate stories were used in this analysis. Any analysis of this kind will of course be subject to bias to some extent, such that others may categorise stories differently or their 'emic' interpretation in the specific society may be different. However, as social content was present in ~70% of stories, approximately double that of the second-highest content (natural phenomena; 39%), it is unlikely that any small changes in categorisation would greatly influence our conclusion that one of the functions of stories in hunter-gatherer societies may be to coordinate social behaviour.

**Ethics**. Ethical clearance was granted by the University College London Ethics Committee (UCL Ethics code 3086/003). Fieldwork permission was granted by local government units, including the Mayors of the Municipalities visited and from the Department of Environment and Natural Resources (DENR) as the research took place in a protected area. Each Agta community agreed to participate and informed consent was obtained from all individuals.

**Data availability**. The data that support the findings of this study are available from the authors upon reasonable request.

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

## Acknowledgements

This project was funded by the Leverhulme Trust grant RP2011-R-045 to A.B.M. and R.M. R.M. also received funding from European Research Council Advanced Grant AdG 249347. We thank R. Schlaepfer for fieldwork support. We also thank our assistants in the Philippines, as well as the Agta communities.

## Author contributions

A.B.M. conceived the project. D.S., A.B.M., R.M. and L.V. designed the experiments. D.S. performed the experiments. D.S. and K.M. collected reputational data. P.S. and A.B.M. collected the data on Agta stories. M.N. and L.A. translated the Agta stories. D.S. collected data on stories from other hunter-gatherer societies. D.S., A.B.M., M.D., K.M. and A.E.P. collected the demographic data. D.S., L.V. and A.B.M. analysed the data. D.S., A.B.M. and L.V. wrote the manuscript. All authors contributed substantially to revisions, analyses and gave final approval for publication.

## Additional information

**Competing interests:** The authors declare no competing financial interests.

