## [Peer Review File · Nature Communications]

Reviewers' comments:

Reviewer #1 (Remarks to the Author):

- ♣ What are the main claims of the paper and how significant are they?

The paper aims to show the beneficial impact of storytelling on a community's ability to cooperate. It also argues that individual ability in storytelling may have been favoured by natural selection, as is proven by the fact that skilled story-teller enjoy larger average offspring than others. To substantiate these claims, the article studies 18 camps of hunter-gatherers from the Philippines. Since these societies resemble in their practices and social patterns of aggregation ancient human societies, I agree with the authors that studying current nomadic societies can shed light on human evolutionary history. The topic of this paper is very important to understand the origins of human cooperation.

- ♣ Who will be interested in reading the paper, and why?

The paper is likely to be of interest for a broad range of scholars, including evolutionary biologists, anthropologists, sociologists and all social scientists interested in human cooperation.

- ♣ Are the claims novel? If not, which published papers compromise novelty? How does the paper stand out from others in its field?

Although the claim that story-telling has been instrumental to the evolution of human cooperation is not new, as the authors acknowledge, their paper is novel for their attempt to put this thesis to an empirical test. The use of experimental techniques is particularly appropriate in this respect to avoid possible confounds that may arise with other methodologies.

- ♣ Are the claims convincing? If not, what further evidence is needed?

I regret to say that I do not find the claims convincing. Below are my main concerns:

1) Causality

34-37: We also show that the presence of good storytellers increases levels of cooperation in camps. Skilled storytellers gain direct fitness benefits in return as they have higher reproductive success and are preferred as camp-mates.

39-40: We conclude that storytelling is an important adaptation to organise cooperative systems in hunter-gatherer societies .

40-42: These stories may have preceded the evolution of other, more complex, fictions promoting cooperation in large-scale societies, such as moralistic high-gods⁷, nation states and political ideologies⁵ .

80-81: [...] we found that camps with a greater proportion of skilled storytellers were associated with increased levels of cooperation.

91-93: The regression coefficient implies that a 1% increase in nominations of good

storytellers increases donations by 2.2 percentage points. This association is consistent with skilled storytellers spreading cooperative norms and promoting cooperation in camps. 95-96: For example, more cooperative camps may tell a greater number of stories, perhaps because they are more socially cohesive

The evidence presented by the authors is purely correlational, but in several parts of the paper (particularly in the introduction) the authors claim causality. The authors are aware that alternative explanations to their claim are possible (see lines 95-96). Quite obviously, story-telling may be a by-product of cooperation, as more cooperative camps may simply facilitate convivial activities. Arguably, camps whose members do not cooperate with each other in real life would also find less interest in coming together in convivial activities. The fact that the measure of storytelling ability is self-reported – rather than being measured on objective grounds – may also bias people from more cooperative camps to praise other camp members. I think that that the topic of this paper is very interesting even if causality cannot be proven. But the authors should avoid implying causality. They should be clear on the theory (or “pathway”) that they want to test, and say that the evidence provided is consistent with this theory. Or, in other words, the evidence they provide does not falsify the theory. The claim at line 40-42 is particularly speculative, and it can be better substantiated.

2) Measurement of storytelling abilities

82-84: We measured individual reputations for storytelling by asking people (n = 297, over 18 camps in two municipalities) to name the best storytellers in camp (we imposed no lower or upper limit on the number of nominations; see Methods).

Although the specific empirical evidence that the authors provide is very convincing and robust, I am concerned that the authors only focus on the number of skilled story-tellers, rather than on the amount of time spent in communal activities involving storytelling. It actually seems to me that the presence of “skilled” story-tellers is of secondary importance to the actual occurrence of story-telling in a community. In fact, the presence of individuals who monopolise conversations in communal talks might even be detrimental to the development of a sense of communal belonging, which is important to foster cooperative attitudes. The inclusion of gossiping in storytelling (on which I comment below) is in this sense problematic because it is questionable that gossiping requires a specific skill in storytelling – at least inasmuch as gossiping entails reporting of facts unknown to the listener. The authors should in my opinion make a better work of justifying theoretically the use of this empirical construct.

I find it rather puzzling that authors asked to report the names of the “best” storytellers in the camp. This hampers comparison between camps. Suppose camp A has 5 very good story-tellers who are clearly better than others, while camp B has 5 mediocre story-tellers who are nonetheless better than any other in the camp. The measure used by the authors would then assign the same ranking to the two camps. A much better way to proceed would have been, in my opinion, to rank the storytelling abilities of other members of the camp on an objective scale – from very good to not good. This would have warranted better between-camp comparison.

3) Definition of storytelling

219-222: 'Storytelling' is defined loosely here to encompass a spectrum of narrative forms, from 'ritualised' storytelling, often in larger groups and accompanied by cosmological or religious content, to less-structured storytelling, in the form of gossip, jokes or regaling the day's events among smaller groups.

I find the conflation of practices such as gossip and 'ritualised' story-telling into one general construct inappropriate. Although it is true that both practices may have a beneficial effect on cooperation, the mechanisms whereby this is possible are very different, thus making it difficult to interpret the empirical results. Gossiping puts one's social image under social scrutiny, thus turning cooperation in one's self-interest to avoid the possibility of retaliation by non-cooperators. Ritualised story-telling arguably strengthens the community's social bonds, thus replacing individual interests with collective interests, and transmits information on social norms. In fact, most of the literature on story-telling remarks the differences between gossiping and ritualised "storytelling". The latter may be "transcendental" in character, while the former is mundane. In her pioneering work, Wiessner (2014, cited by the authors) notes that gossiping normally occurs during the day, while folktale and ritualized storytelling occurs at night. This also entails that gossiping occurs among a lower number of people than ritualised storytelling. Wiessner (2014: 14027) writes: "Day talk centered on practicalities and sanctioning gossip; firelit activities centered on conversations that evoked the imagination, helped people remember and understand others in their external networks, healed rifts of the day, and conveyed information about cultural institutions that generate regularity of behavior and corresponding trust".

4) The experimental game measures altruism but not propensity to cooperate

85-87: We also asked 290 Agta adults from these camps to play an experimental resource allocation game in which subjects could either keep or share any desired fraction of a resource.

The authors use a modified version of a Dictator Game as a measure of cooperation. In a Dictator Game a player is asked to divide an endowment of resources (rice in this case) between him/herself and another person. In this case there are many possible recipients of the agent's decision, but the nature of the game remains the same. My concern is that this game is not a measure of cooperation, strictly speaking, but rather of altruism. A key requirement of cooperation is that cooperative actions bring about benefits to others that exceed the costs for the agent (see e.e. Nowak, 2006, cited by the authors). Mutual cooperation thus permits net gains in resources. A Dictator Game does not involve any creation of additional surplus as a result of the agents' choices. It merely consists in a transfer of resources from one agent to another. Clearly the propensity to share resources with others is a signal of pro-social disposition that are also at the base of propensity to cooperate. Nonetheless, the two notions are different. It is therefore unfortunate that the authors did not use a game such as a Prisoner's Dilemma or even a "Helping game", which can allow for efficiency gains.

I am also not clear as to whether the subject of one decision may have also been the recipient of somebody else's decision. The authors do not include experimental instructions therefore I could not check. It is a good practice that Dictators and Recipients are

exclusively assigned to one of the two roles, in order to tell apart altruism from reciprocity. It is not clear to me whether the relationship was potentially reciprocal or not, and whether and how this important aspect was communicated to participants.

5) Interpretation of indication of skilled storyteller as preferable camp companions

96-100: Therefore, if storytelling plays a functional role in promoting cooperation, we predict that skilled storytellers would be preferred as social partners. In contrast, if storytelling is only a consequence or by-product of cooperation, preferred social partners are likely to be chosen on the basis of other characteristics, such as foraging skills or medicinal knowledge .

I think this claim is rather unsubstantiated. Storytellers may be chosen simply because they are good entertainers, rather than for their effects on camp-level cooperation.

6) Interpretation of stories

33-34: Agta stories convey messages relevant to coordinating group behaviour in a foraging ecology, such as cooperation, sex equality and social egalitarianism

65-69: All stories conveyed norms and principles regulating cooperation and social behaviour, specifically sex equality ('The sun and the moon'), social egalitarianism and friendship ('The wild pig and the seacow'), group cooperation ('The monkey and the giant'), and group identity and social acceptance ('The winged ant').

385-7: Not all stories obtained from these populations concerned social behaviour. Many stories not described here concern cosmological content, with seemingly little social relevance.

511: There is a dispute between the sun (male) and the moon (female) to illuminate the sky. After a fight, where the moon proves to be as strong as the sun, they agree in sharing the duty - one during the day and the other during the night.

There are always going to be elements of arbitrariness, and possibly ethnocentrism, in the construal of the stories. But I found some of the authors' claims not truly convincing. First of all, I think the authors should quantify the relative frequency of stories with cosmological content from those affecting social relations (see lines 385-7). If the latter are hardly ever told, then the effect of storytelling on cooperation is limited. Some of the interpretations are also disputable. In the first reported (see line 511), is the attribution of sun and moon to male and female genders the authors' interpretation, or is this characteristic stressed in the original story? In the "Wild pig and seacow" story, construal in terms of social equality is doubtful.

♣ Are there other experiments or work that would strengthen the paper further?

The authors' main claim is that able story-telling causes cooperation to increase. As I argued above, causality is hard to be attained in this setting. Nevertheless, the authors may consider carrying out a "longitudinal" work – meaning following camps over time – and recording how patterns of cooperation change as change in camp composition and

occurrence of storytelling occurs. A storyteller ability is probably a characteristic that stays constant over time. Nevertheless, the frequency and duration of storytelling may vary depending on factors such as seasonality and weather events (cold or rainy seasons probably hampers meetings and thus disfavours storytelling), as well as the periodic disbanding and re-composition of camps. The authors may exploit the natural variability in these variables (which is exogenous in the first two cases) and try to attain a direct causal link between frequency of storytelling "events" and cooperation. For instance, they may be able to show that as skilled storytellers move from one camp to the other, cooperation levels increase.

An alternative strategy that the authors may follow to substantiate their causality claims is through priming. In the treatment condition, participants may attend some storytelling session by skilled storytellers before taking part in the cooperation game. In the control condition, such storytelling session would be absent (or even better, substituted by some individual activity, such as reading an essay or listening to a speaker in isolation from other people). The expected effect is that cooperation levels should be higher in the treatment condition.

♣ Is the manuscript clearly written? If not, how could it be made more clear or accessible to nonspecialists?

The manuscript is written with accurate language and is broadly accessible. I would advise the authors to make all the necessary supporting and descriptive material accessible in this same paper. A reader cannot be expected to read a previously published paper by the authors to obtain relevant information.

♣ Is the statistical analysis of the data sound, and does it conform to the journal's guidelines?

The analysis is sound. However the authors should specify the number of observations in each regression / test. They should also explain what a z-score is. Why are there 6,534 dyads?

Below some other requests for clarification.

264-266: To allow all prestige measures across all individuals to be compared in the same model, all women were assigned a '0' for hunting and fishing, while all men were given a '0' for tuber-gathering.

This seems questionable.

514-515: Table 2: Models assessing the likelihood of skilled storytellers being selected in a 'camp-mate' network, using a logistic GEE regression ($n = 291$, dyads = 6,534).

How extensive is the network considered? Would for instance people from one municipality be allowed to evaluate links with people from the other municipality?

572: Extended Data Table 3
Why proximity has a negative sign?

Reviewer #2 (Remarks to the Author):

This paper should be of interest to all researchers concerned with the evolution of human cooperation. Many researchers have argued that language at least facilitates, and perhaps is even required, for the extraordinary degree of collective action (group cooperation) in our species. But very few have focused on narrative (storytelling) as done here, and I cannot think of any cases where the role of storytelling in enhancing cooperation has been subject to careful empirical test. The paper very clearly lays out the claims, the evidence, and the methods used to collect this evidence.

The paper makes three central claims: 1) storytelling promotes group cooperation; 2) storytellers gain fitness benefits; and (the broadest and most significant) 3) the "results provide a plausible pathway by which a group-beneficial behaviour such as storytelling might have evolved through individual-level selection".

The evidence presented for these claims can be summarized as follows (numbers matching those I list above for the claims):

- 1) Agta stories have content promoting norms of cooperation and egalitarianism. The presence of good storytellers (as measured by free-listing nominations) increases camp cooperation (as measured using a modified Dictator game).
- 2) Storytellers are preferred as co-residents (as measured by the correlation between storytelling reputation and free-listing nominations of preferred camp mates). Storytellers have higher reproductive success on average.
- 3) Given above, there is no free-rider problem with storytelling that would require group-level selection.

There are some issues with the evidence (again, numbering corresponds to above):

- 1) The sample of stories is very small ($n=4$); however, the paper attempts to blunt this problem with reference to stories from other hunter-gatherer societies (see "Extended Data Table 1"). There is no formal analysis of content -- the claim that stories promote group cooperation is impressionistic. No information is provided on individual mobility between camps; if it is high (which is known to be the case in many but not all other hunter-gatherer societies), and includes storytellers, it would weaken the case for seeing the correlation between presence of good storytellers and increased camp cooperativeness. Finally, the Dictator game is not on the face of it a good way to measure willingness to engage in collective action (group-level cooperation), as it provides the same score for giving $x\%$ to one other player as $x\%/n$ to n other players.
- 2) The link between storytelling reputation and camp-mate preference is purely correlational, making it difficult to rule out alternative explanations (as is also the case with the evidence for claim 1). However, the researchers did control for many other variables that plausibly could drive this correlation, which increases our confidence that the claim is supported.

3) The evidence that storytellers have higher average RS does address the free-rider problem with regard to why be a storyteller rather than a passive listener. However, it does not answer the question of why others should provide hypothesized "social support" to storytellers (at presumably some personal cost), so the claim that individual-level benefits account for the system is incomplete.

A minor quibble: Refs 20 & 21 are too general to support the claim that storytelling "is a costly behaviour requiring an input of time and energy into practice, performance and cognitive processing"; it could well be that some individuals just have "the gift" and it comes easily (and of course brings social rewards). However, this is not critical for evaluating the key claims listed above.

In sum, this is an impressive piece of research. Although I am not qualified to judge the statistical analyses in detail, the methods are laid out clearly, and seem careful enough, that I have no technical concerns about the analyses. Despite some issues with the degree to which the evidence adequately tests the claims (described above), I do think the paper is worthy of publication, given some revisions.

Reviewer #3 (Remarks to the Author):

Summary of Article

The major claims of this paper are that storytelling and fiction are an adaptation whose function is to promote cooperation in hunter-gatherer societies. In support of this claim, they present evidence that: (1) the Agta prefer skilled storytellers as group members over other skill specialists such as accomplished fishermen and hunters; (2) good storytellers have higher reproductive success; (3) Agta stories communicate information relevant to cooperation and egalitarianism; and (4) the presence of good storytellers increases cooperation in camps.

Methods

The investigators collected four Agta stories that elders "normally tell children and each other" (63). Each of these stories communicated norms and principles regulating social behavior: one story conveyed information about sex equality; one conveyed information about social egalitarianism; one conveyed information about group cooperation; and one conveyed information about group identity. This information includes an emphasis on the benefits of cooperation, examples of punishment for violating behavioral norms, and references to the use of reverse dominance hierarchies to prevent monopolization of power. They note that these themes occur in other hunter-gatherer groups (Extended Data Table 1): their evidence consists of a total of 11 stories from 6 other Southeast Asian hunter-gatherer groups (including another Agta population).

As the authors note, the higher degree of cooperativeness (as measured by the Dictator

Game) found in groups with more storytellers is consistent with skilled storytellers spreading cultural norms, but is not conclusive because other explanations for these results are possible. Similarly, the findings that storytellers have higher reproductive success and that group members prefer skilled storytellers do not directly speak to the question of whether storytelling/fiction is an adaptation. As for the study sample, the size (four stories from the Agta) is adequate: the fact that these themes occur in stories that are popular in Agta culture is unequivocal evidence that people tell stories about these subjects. However, because they are limited to the same geographical region, neither the Agta data nor the comparative data provides compelling evidence for the larger claim that storytelling is an adaptation and that it evolved to promote cooperation.

Content

(23) "Storytelling occurs spontaneously in childhood"

The reference cited in support of this claim (Boyd 2009) is unsuitable. Boyd is a literary scholar, not a developmental psychologist, has no training in developmental psychology, and does not do quantitative research on the acquisition of narrative by children. Much more compelling evidence comes from the work of Pitcher & Prelinger (1963) and Sutton-Smith (1981), who collected large samples of stories generated by preschool and elementary school children. For a review of this research, see Scalise Sugiyama 2009.

(24) "has the power to increase empathy towards others"

This conclusion is premature and only weakly supported. The Djikic et al. (2009) study cited by the authors shows that a fictional story arouses more emotion than a "court document meant to represent an ostensible divorce proceeding," but does not show that reading literature increases empathy. Indeed, Djikic et al.'s experiment does not test directly for empathy. Moreover, subjects were administered the Emotions Checklist immediately after reading either the Chekov story or the "court document," which provides no information about any hypothesized long-term effects of reading literature. Kidd & Castano (2013) similarly found that, compared to reading nonfiction, popular fiction, or nothing at all, reading literary fiction temporarily enhances theory of mind, as measured using the Reading the Mind in the Eyes Test. Again, however, their study does not speak to the long-term effects of reading literature vis-à-vis ToM abilities. Furthermore, Panero et al. (2016) recently failed to replicate Kidd & Castano's (2013) findings using three independent research groups and a total of 792 participants. Other studies have found that the effects of reading fiction are moderated by individual difference variables such as transportation into the story, affective empathy, and openness to experience (e.g., Bal & Veltkamp 2013, Johnson 2012, Djikic et al. 2013).

(28-29) "Despite its undeniable importance, little attention has been given to understanding the function and evolution of human storytelling"

Obviously, this is a matter of opinion, but there is certainly more research on this subject than the authors indicate. See, for example Minc 1986; Scalise Sugiyama 1996, 2005, 2008, 2011, 2012, 2014; Sobel & Bettles 2000; Steen & Owens 2001; Tooby & Cosmides 2001.

(29-30) "Here we propose that fiction and storytelling promote cooperation in hunter-gatherer groups"

The hypothesis that storytelling is used to promote cooperation is not novel. See Minc 1986; Sobel & Bettles 2000; Scalise Sugiyama 2011. Also, fiction and storytelling are not the same thing (Tooby & Cosmides 2001; Scalise Sugiyama 2012). There is a large literature in psychology on fiction (a.k.a., pretense), which is widely viewed as a manifestation of the capacity for reasoning counterfactually (a.k.a. conditional reasoning; Leslie 1987; Cosmides & Tooby 2000; Onishi et al. 2007). This, in turn, is the foundation of perspective taking-- i.e., imagining the environment from different spatial, temporal, or psychological perspectives (see, e.g., Schacter et al. 2007), the latter of which is typically referred to as theory of mind.

(39-40) "We conclude that storytelling is an important adaptation to organise cooperative systems in hunter-gatherer societies"

Earlier in the paper, the authors include fiction in this claim—"fiction and storytelling promote cooperation in hunter-gatherer groups" (29-30). The claim needs to be clarified (are both fiction and storytelling adaptations, or only storytelling?), especially since fiction and storytelling are not the same thing (see previous comment). Additionally, in order to claim that storytelling is an adaptation, the authors need to define it cognitively (story grammar research is a good place to start—e.g., Rumelhart 1975; Schank 1975; Thorndyke 1977). Their provisional definition ("an account of a sequence of events in the order in which they occurred to make a point") is incomplete and cognitively nebulous. Finally, the authors must demonstrate that there is an adaptive problem to be solved, and that storytelling exhibits design aimed at solving this problem. The present article does not discuss design, nor does it demonstrate that storytelling performs a function essential to cooperation. Moreover, the article glosses over that fact that forager oral storytelling transmits other types of information besides social norms: "Not all stories obtained from these populations concerned social behaviour. Many stories not described here concern cosmological content, with seemingly little social relevance. Examples include Batek origins of the cosmos, Maniq origins of night and day (originating from a snake continuously eating and regurgitating its tail), and Andamanese origins of fire (see references in Extended Data Table 1). Their existence does not diminish the importance of stories in facilitating cooperation, but merely highlights that stories can perform other functions, such as disseminating fitness-relevant information" (385-391). As the authors concede, storytelling transmits other types of adaptively relevant knowledge. For example, one common theme in forager oral tradition is dangerous animals; thus, to make the case that storytelling evolved

to "organise cooperation," one must demonstrate that storytelling did not evolve in response to other adaptive problems, such as predator avoidance. Humans cooperate all the time without storytelling, which belies the claim that storytelling evolved and is designed expressly to facilitate cooperation. For a discussion of storytelling vis-à-vis cooperation, see Scalise Sugiyama's (2008:31-32) article about tricksters, as well as Scalise Sugiyama 2011 (12-14) on storytelling and the broadcasting of social norms.

(70-73) "while also exemplifying various mechanisms of social norm enforcement, such as emphasising the benefits to cooperation over competition, examples of punishment for breaking norms, and reverse dominance hierarchies to prevent individual accumulation of power"

For a discussion of the use of storytelling for intentional leveling and social sanctioning, see Scalise Sugiyama 2011, 2012.

(125-126) "what would be the individual benefit for storytellers"

This question has been addressed and answered elsewhere. See Scalise Sugiyama 2005, 2011, 2012.

(142-143) "It is possible that by performing an important social function, skilled storytellers receive increased social support from others"

This idea has been proposed elsewhere: see Scalise Sugiyama & Sugiyama 2003.

Recommendation

Of value in this study is the evidence it provides from 7 different Southeast Asian hunter-gatherer groups that storytelling transmits information related to social norms. This finding needs to be integrated into the large body of research on the function (evolved or otherwise) that storytelling serves in preliterate, small-scale societies, most of which the authors do not cite. The same goes for the authors' claims regarding storytelling and empathy: their review of this literature is thin and problematic. Finally, the study does not provide convincing theoretical or quantitative evidence in support of its claim that storytelling/fiction is an adaptation or that its function is to "organize cooperation." For these reasons, I do not find the article suitable for publication in Nature Communications.

Reviewers' comments:

Reviewer #1 (Remarks to the Author):

We thank the reviewer for their insightful and detailed comments, which we have responded to and included as fully as possible in our revised manuscript. We have addressed all of the reviewer's major comments, including issues regarding causality, measurement of storytelling skill, definitions of storytelling and a content analysis demonstrating the importance of stories to coordinate hunter-gatherer social behaviour, as well as other issues raised. We believe that we have replied to all the reviewer's comments, and that our analyses support the main arguments of our article, namely that hunter-gatherer stories appear designed to coordinate social and cooperative behaviour, that the presence of skilled storytellers is associated with increased cooperation, and that skilled storytellers receive fitness benefits, both in terms of enhanced social support and increased reproductive success. These findings are consistent with our hypothesis that stories evolved to coordinate social behaviour and promote cooperation among hunter-gatherers, as well as providing a mechanism by which group-beneficial behaviour (storytelling) can evolve via individual-level selection. We believe that these conclusions are novel, supported by rigorous analyses, and of sufficient interest to a wide range of researchers to merit publication in Nature Communications. We hope that these revisions address the issues raised and the reviewer agrees that our paper represents an important and relevant contribution to the study of human behaviour and will consider our manuscript for publication. The reviewer's comments are in black font with our replies in red font.

♣ What are the main claims of the paper and how significant are they?

The paper aims to show the beneficial impact of storytelling on a community's ability to cooperate. It also argues that individual ability in storytelling may have been favoured by natural selection, as is proven by the fact that skilled story-teller enjoy larger average offspring than others. To substantiate these claims, the article studies 18 camps of hunter-gatherers from the Philippines. Since these societies resemble in their practices and social patterns of aggregation ancient human societies, I agree with the authors that studying current nomadic societies can shed light on human evolutionary history. The topic of this paper is very important to understand the origins of human cooperation.

♣ Who will be interested in reading the paper, and why?

The paper is likely to be of interest for a broad range of scholars, including evolutionary biologists, anthropologists, sociologists and all social scientists interested in human cooperation.

♣ Are the claims novel? If not, which published papers compromise novelty? How does the paper stand out from others in its field?

Although the claim that story-telling has been instrumental to the evolution of human cooperation is not new, as the authors acknowledge, their paper is novel for their attempt to put this thesis to an empirical test. The use of experimental techniques is particularly appropriate in this respect to avoid possible confounds that may arise with other methodologies.

♣ Are the claims convincing? If not, what further evidence is needed?

I regret to say that I do not find the claims convincing. Below are my main concerns:

1) Causality

34-37: We also show that the presence of good storytellers increases levels of cooperation in camps. Skilled storytellers gain direct fitness benefits in return as they have higher reproductive success and are preferred as camp-mates.

39-40: We conclude that storytelling is an important adaptation to organise cooperative systems in hunter-gatherer societies .

40-42: These stories may have preceded the evolution of other, more complex, fictions promoting cooperation in large-scale societies, such as moralistic high-gods⁷, nation states and political ideologies⁵ .

80-81: [...] we found that camps with a greater proportion of skilled storytellers were associated with increased levels of cooperation.

91-93: The regression coefficient implies that a 1% increase in nominations of good storytellers increases donations by 2.2 percentage points. This association is consistent with skilled storytellers spreading cooperative norms and promoting cooperation in camps.

95-96: For example, more cooperative camps may tell a greater number of stories, perhaps because they are more socially cohesive

The evidence presented by the authors is purely correlational, but in several parts of the paper (particularly in the introduction) the authors claim causality. The authors are aware that alternative explanations to their claim are possible (see lines 95-96). Quite obviously, story-telling may be a by-product of cooperation, as more cooperative camps may simply facilitate convivial activities. Arguably, camps whose members do not cooperate with each other in real life would also find less interest in coming together in convivial activities. The fact that the measure of storytelling ability is self-reported – rather than being measured on objective grounds - may also bias people from more cooperative camps to praise other camp members. I think that that the topic of this paper is very interesting even if causality cannot be proven. But the authors should avoid implying causality. They should be clear on the theory (or “pathway”) that they want to test, and

say that the evidence provided is consistent with this theory. Or, in other words, the evidence they provide does not falsify the theory. The claim at line 40-42 is particularly speculative, and it can be better substantiated.

We agree that our analyses are correlational, and have now updated the language in several places as noted by the reviewer so as not to imply causality; rather, our results are consistent with the theory we have proposed. Specifically, we now state that:

30-31: [...] the presence of good storytellers *is associated with* increased levels of cooperation.

35-36: [...] storytelling *may be* an important mechanism to organise cooperative systems in hunter-gatherer societies.

111-113: [...] a 1% increase in nominations of good storytellers *was associated with* an increase in donations by 2.2 percentage points.

179-180: We conclude that storytelling *may be* an adaptation to organise cooperative systems in hunter-gatherer societies [...].

220-221: In sum, we argue that storytelling *may be* an adaptation to organise cooperation in hunter-gatherers [...].

224-226: From simple storytelling to complex religion and later formal institutions such as nation states, the evolution of storytelling *may have* been pivotal in organising and promoting human cooperation.

We have also included a new section which specifically addresses these concerns regarding causality (lines 199-202): 'The evidence provided here is consistent with the theory that storytelling acts as a mechanism to coordinate group behaviour and promote cooperation. However, these findings are largely correlational and further studies are required to conclusively demonstrate that storytelling performs a causal role in facilitating cooperative behaviour.'

Additionally, we agree that although our previous claim that 'these stories may have preceded the evolution of other, more complex, fictions promoting cooperation in large-scale societies, such as moralistic high-gods, nation states and political ideologies' (lines 40-42 in original submission) is rather speculative, it is a logical extension of our theory applied to larger agricultural societies. Indeed, others have made similar arguments for the necessity of such stories in facilitating large-scale cooperation (Harari, 2011) and recent research regarding the presence of high-gods associated with increased levels of cooperation is also consistent with this interpretation (Purzycki et al., 2016). However, we have now removed this claim from the abstract and toned down our claims throughout the paper.

2) Measurement of storytelling abilities

82-84: We measured individual reputations for storytelling by asking people (n = 297, over 18 camps in two municipalities) to name the best storytellers in camp (we imposed no lower or upper limit on the number of nominations; see Methods).

Although the specific empirical evidence that the authors provide is very convincing and robust, I am concerned that the authors only focus on the number of skilled story-tellers, rather than on the amount of time spent in communal activities involving storytelling. It actually seems to me that the presence of “skilled” story-tellers is of secondary importance to the actual occurrence of story-telling in a community. In fact, the presence of individuals who monopolise conversations in communal talks might even be detrimental to the development of a sense of communal belonging, which is important to foster cooperative attitudes. The inclusion of gossiping in storytelling (on which I comment below) is in this sense problematic because it is questionable that gossiping requires a specific skill in storytelling – at least inasmuch as gossiping entails reporting of facts unknown to the listener. The authors should in my opinion make a better work of justifying theoretically the use of this empirical construct.

I find it rather puzzling that authors asked to report the names of the “best” storytellers in the camp. This hampers comparison between camps. Suppose camp A has 5 very good story-tellers who are clearly better than others, while camp B has 5 mediocre story-tellers who are nonetheless better than any other in the camp. The measure used by the authors would then assign the same ranking to the two camps. A much better way to proceed would have been, in my opinion, to rank the storytelling abilities of other members of the camp on an objective scale – from very good to not good. This would have warranted better between-camp comparison.

While we acknowledge that our measure of proportion of storytellers in camp is a proxy for the impact that storytellers have on their camp, several other studies have used similar proxies for inferring foraging skill and investment in group activities (Marlowe, 1999; Macfarlan & Lyle, 2015). Indeed, using the same protocols we show that individuals with a greater reputation for fishing were associated with both greater returns per hour and total calories obtained from fishing (lines 342-343). Nevertheless, we agree that future studies should aim to employ less distal measures, such as actual time spent in storytelling activities to further explore our conclusions. We therefore included a new sentence in our discussion (lines 211-214) stating that ‘[...] although we demonstrate an association between the presence of skilled storytellers in camp and levels of cooperation, future studies may benefit from utilising more direct indices of storytelling, such as the amount of storytelling in a camp, to assess this putative link in greater detail.’

Given the amount of data collected, and the validation of previous studies using similar designs (Marlowe, 1999; Macfarlan & Lyle, 2015), we decided to ask individuals to name the best storytellers (in addition to other reputational domains) as this would permit greater breadth of data collection. While other methods are possible (as the reviewer

suggested), these would require greater time investment, which would necessarily detract from other aspects of data collection. Additionally, even with such 'objective scales' of storytelling it is also possible that the same potential issues of comparisons between camps would exist. For instance, if individuals were asked to categorise others as 'very good' to 'not good' storytellers, individuals may still have answered relative to their camp. Irrespective of this possibility, we have conducted new analyses which demonstrate substantial variability in the frequency of individuals who nominated no-one as a skilled storyteller between camps (as individuals had the option of nominating as few or as many camp-mates as they wished). This suggests that nominations were not merely relative to others in camp, but rather reflect absolute differences in storytelling skill. Between-camp comparisons using this data are therefore valid using this method. This has been updated in our manuscript:

284-294: One potential issue with this methodology is that it focuses on within-camp comparisons of storytelling ability, such that storytelling skill may be relative to each camp, rather than absolute for the population, which may hinder between-camp comparisons. However, this is unlikely to have occurred as the average proportion of nominations for each individual per camp ranged from 0.03 to 0.21 (see 'Resource Allocation Game' section below), indicative of substantial between-camp variation in the number of nominations for storytelling ability. Additionally, the proportion of individuals who nominated no-one varied considerably between camps (from 0% to 62.5%). A chi-square test of independence reported that the frequency of individuals nominating no-one varied significantly between camps ($\chi^2(17, n=297)=82.28, p<0.001$). Together, these suggest that storyteller nominations were not simply relative to camp-mates, but rather reflect absolute differences in skill.

3) Definition of storytelling

219-222: 'Storytelling' is defined loosely here to encompass a spectrum of narrative forms, from 'ritualised' storytelling, often in larger groups and accompanied by cosmological or religious content, to less-structured storytelling, in the form of gossip, jokes or regaling the day's events among smaller groups.

I find the conflation of practices such as gossip and 'ritualised' story-telling into one general construct inappropriate. Although it is true that both practices may have a beneficial effect on cooperation, the mechanisms whereby this is possible are very different, thus making it difficult to interpret the empirical results. Gossiping puts one's social image under social scrutiny, thus turning cooperation in one's self-interest to avoid the possibility of retaliation by non-cooperators. Ritualised story-telling arguably strengthens the community's social bonds, thus replacing individual interests with collective interests, and transmits information on social norms. In fact, most of the literature on story-telling remarks the differences between gossiping and ritualised "storytelling". The latter may be "transcendental" in character, while the former is mundane. In her pioneering work, Wiessner (2014, cited by the authors) notes that gossiping normally occurs during the day, while folktale and ritualized storytelling occurs at night. This also entails that gossiping occurs among a lower number of people than ritualised storytelling. Wiessner (2014: 14027) writes: "Day talk centered on practicalities

and sanctioning gossip; firelit activities centered on conversations that evoked the imagination, helped people remember and understand others in their external networks, healed rifts of the day, and conveyed information about cultural institutions that generate regularity of behavior and corresponding trust”.

We agree with the reviewer that the distinction between ‘storytelling’ and ‘gossip’ is important. We have therefore removed our discussion on gossip and focussed on ritualised and other forms of storytelling. Our manuscript has now been altered to read:

252-267: ‘Storytelling’ is defined loosely here to encompass a spectrum of narrative forms, from ‘ritualised’ storytelling, often in larger groups and accompanied by cosmological or religious content, to less-structured storytelling as a part of everyday conversation among smaller groups. Cutting across these contextual differences, a story can be broadly defined as “an account of a sequence of events in the order in which they occurred to make a point”⁵⁰. A story can also be defined by its components, and several lines of evidence have converged from literary theory and cognitive psychology which suggest that narratives consist of: character, setting, events, causal connections and resolution (for a review see²²). These are inclusive definitions of ‘storytelling’, which encompass both ritualised or fictional stories, as well as many aspects of everyday conversation which also contain non-fictional narratives, such as jokes, anecdotes or details of previous experiences. Although much research focuses on fictional stories²⁷, the importance of non-fictional narratives should not be overlooked. For instance, in her work with the Ju/’hoansi, Wiessner²³ provides details of several stories, all of which are non-fictional, which broadcast social norms concerning issues such as sex, marriage, sharing obligations and norm-breakers (see also Supplementary Table 1 for other examples of non-fictional stories, such as by the Batak).

4) The experimental game measures altruism but not propensity to cooperate

85-87: We also asked 290 Agta adults from these camps to play an experimental resource allocation game in which subjects could either keep or share any desired fraction of a resource.

The authors use a modified version of a Dictator Game as a measure of cooperation. In a Dictator Game a player is asked to divide an endowment of resources (rice in this case) between him/herself and another person. In this case there are many possible recipients of the agent’s decision, but the nature of the game remains the same. My concern is that this game is not a measure of cooperation, strictly speaking, but rather of altruism. A key requirement of cooperation is that cooperative actions bring about benefits to others that exceed the costs for the agent (see e.e. Nowak, 2006, cited by the authors). Mutual cooperation thus permits net gains in resources. A Dictator Game does not involve any creation of additional surplus as a result of the agents’ choices. It merely consists in a transfer of resources from one agent to another. Clearly the propensity to share resources with others is a signal of pro-social disposition that are also at the base of

propensity to cooperate.

Nonetheless, the two notions are different. It is therefore unfortunate that the authors did not use a game such as a Prisoner's Dilemma or even a "Helping game", which can allow for efficiency gains.

I am also not clear as to whether the subject of one decision may have also been the recipient of somebody else's decision. The authors do not include experimental instructions therefore I could not check. It is a good practice that Dictators and Recipients are exclusively assigned to one of the two roles, in order to tell apart altruism from reciprocity. It is not clear to me whether the relationship was potentially reciprocal or not, and whether and how this important aspect was communicated to participants.

We agree that cooperation must be beneficial for both parties in the long-term in order to evolve. However, we would disagree that a game which permits short-term 'efficiency gains' is required to demonstrate the occurrence of cooperation. From an evolutionary perspective, cooperation can simply be defined as a 'behaviour which evolved to help others' (West et al., 2007), therefore any behaviour which demonstrates sharing resources with others at a cost to self (such as the one used here) can be described as 'cooperative'. The sharing behaviour observed in these games appears 'altruistic' in the short-term, but may lead to 'efficiency gains' for both parties when measured in terms of increased fitness on a long-term evolutionary time-scale.

Related to this, there is currently a debate over whether behaviour in these economic games is actually altruistic, or rather a misfiring of other mechanisms (such as reciprocity, reputation or experience; Delton et al., 2011; Rand et al., 2014), meaning that behaviour in these games may not be strictly altruistic (i.e., resulting in decreased fitness), but rather reflect other cooperative processes which may be adaptive outside of the experimental context.

Participants were told that the same game would be played with other camp-mates, meaning that the potential for reciprocal nominations was known to all players. As we were interested in assessing cooperation (which can include reciprocity), rather than just prosocial intent, we did not wish to separate 'proposers' and 'recipients', but rather use this game as a model for exploring cooperation among the Agta more widely. We have also updated our Methods section to include additional details about the game protocols to clarify the experimental set-up (line 361-367): 'This non-anonymous game structure was used in order for both levels of cooperative behaviour and patterns of cooperation (i.e., who individuals share with, such as kin, reciprocal partners or storytellers) to be ascertained. This game was designed to be intuitive to grasp for the Agta without the need for complex explanations which may lead to frustration, boredom and misunderstandings, as well as to mimic food-sharing decisions made on a daily basis in this society^{4,32}.'

5) Interpretation of indication of skilled storyteller as preferable camp companions

96-100: Therefore, if storytelling plays a functional role in promoting cooperation, we predict that skilled storytellers would be preferred as social partners. In contrast, if storytelling is only a consequence or by-product of cooperation, preferred social partners are likely to be chosen on the basis of other characteristics, such as foraging skills or medicinal knowledge .

I think this claim is rather unsubstantiated. Storytellers may be chosen simply because they are good entertainers, rather than for their effects on camp-level cooperation.

We concur that this 'entertainment value' perspective is one potential interpretation. However, we have included a new analysis (Supplementary Tables 10 & 11) which demonstrates that skilled storytellers are also more likely to receive resources than non-skilled storytellers in the experimental game. This shows that individuals share with storytellers at a cost to themselves. These patterns of costly cooperation would be unlikely if storytellers played no functional role in society but were solely for 'entertainment value', although this is still a possibility. For details of this additional analysis we refer the reviewer to lines:

170-178: [...] we also demonstrate that skilled storytellers are more likely to be recipients of resource transfers in the experimental game (Supplementary Tables 10 & 11). This suggests that storytellers may be 'rewarded' for their public good by other camp-mates who benefit from the increased cooperation which storytellers may promote, in what may be mutually-beneficial trade-like relationships. Alternatively, people might enjoy listening to stories for other reasons and be paying for the service (this effect could be independent from the function of storytelling in promoting cooperation).

418-423: The resource allocation network employed an identical logistic GEE regression approach using the same methodology and variables as described above [for the camp-mate network], but here using nominations of who individuals distributed resources to in the resource allocation game ($n=290$, dyads=1,312). In this analysis the proportion of resources kept for self was used as a control variable to ensure that patterns of resource distributions were not confounded with overall levels of cooperation.

6) Interpretation of stories

33-34: Agta stories convey messages relevant to coordinating group behaviour in a foraging ecology, such as cooperation, sex equality and social egalitarianism

65-69: All stories conveyed norms and principles regulating cooperation and social behaviour, specifically sex equality ('The sun and the moon'), social egalitarianism and friendship ('The wild pig and the seacow'), group cooperation ('The monkey and the giant'), and group identity and social acceptance ('The winged ant').

385-7: Not all stories obtained from these populations concerned social behaviour. Many stories not described here concern cosmological content, with seemingly little social

relevance.

511: There is a dispute between the sun (male) and the moon (female) to illuminate the sky. After a fight, where the moon proves to be as strong as the sun, they agree in sharing the duty - one during the day and the other during the night.

There are always going to be elements of arbitrariness, and possibly ethnocentrism, in the construal of the stories. But I found some of the authors' claims not truly convincing. First of all, I think the authors should quantify the relative frequency of stories with cosmological content from those affecting social relations (see lines 385-7). If the latter are hardly ever told, then the effect of storytelling on cooperation is limited. Some of the interpretations are also disputable. In the first reported (see line 511), is the attribution of sun and moon to male and female genders the authors' interpretation, or is this characteristic stressed in the original story? In the "Wild pig and seacow" story, construal in terms of social equality is doubtful.

Following the suggestions of the reviewer we have conducted a literature search of stories found in seven forager societies and quantified the frequency of different story types (coding for social, cosmological, resource use, and natural phenomena content). As we now describe in lines 93-97: 'To explore the content of hunter-gatherer stories in greater detail we collected 89 stories over seven different forager societies and coded them according to subject matter (see Methods for further details). Of these stories, ~70% were classified as pertaining to 'social behaviour' (i.e., prescribing social norms or coordinating behavioural expectations), more than any other category (see Supplementary Table 2).' This bolsters our hypothesis that storytelling may have evolved to coordinate social behaviour and facilitate cooperation.

For additional details on this analysis see lines 480-499: 'In addition to the stories concerning social behaviour (Supplementary Table 1), we also performed a content analysis of all stories collected ($n=89$) from these Southeast Asian hunter-gatherer populations in addition to two African forager groups (Ju/'hoansi and BaYaka; Supplementary Table 2). Each story was assessed for different types of content: social (content which prescribed behaviour during interactions with others, such as cooperation, sex equality, norm-breaking, sex roles, punishment and interactions with out-groups); cosmological (content concerning the origins of the earth/universe); natural phenomena (content concerning navigation, animal origins/behaviour, fire and natural disasters/weather); and resource use (content about foraging and resource extraction). Each of these broad themes are common in forager folklore^{15,23,27,41,61}. Many stories were classified along more than one criterion, such as many creation stories combining cosmological and social content (see Supplementary Table 1). Some sources presented several versions of the same basic story; to prevent unnecessary duplication, in these cases only the longest or most elaborate stories were used in this analysis. Any analysis of this kind will of course be subject to bias to some extent, such that others may categorise stories differently or their 'emic' interpretation in the specific society may be different. However, as social content was present in ~70% of stories, approximately

double that of the second-highest content (natural phenomena; 36%), it is unlikely that any small changes in categorisation would greatly influence our conclusion that stories in hunter-gatherer societies appear to function to coordinate social behaviour.'

Regarding interpretation of the Agta stories specifically, the male/female associations of the sun and the moon were given by the Agta (and similar associations are found in many other Southeast Asian hunter-gatherer societies). We do agree that there will often be disagreements over interpretation of stories (especially without knowing how they are interpreted from an 'emic' perspective by the societies themselves, as stated above). Although we believe that the story 'the wild pig and the seacow' does reflect social equality, we have removed this from Table 1 and kept 'cooperation' and 'friendship', as this makes little difference to our argument as it still reflects social behaviour.

♣ Are there other experiments or work that would strengthen the paper further?

The authors' main claim is that able story-telling causes cooperation to increase. As I argued above, causality is hard to be attained in this setting. Nevertheless, the authors may consider carrying out a "longitudinal" work – meaning following camps over time – and recording how patterns of cooperation change as change in camp composition and occurrence of storytelling occurs. A storyteller ability is probably a characteristic that stays constant over time. Nevertheless, the frequency and duration of storytelling may vary depending on factors such as seasonality and weather events (cold or rainy seasons probably hampers meetings and thus disfavours storytelling), as well as the periodic disbanding and re-composition of camps. The authors may exploit the natural variability in these variables (which is exogenous in the first two cases) and try to attain a direct causal link between frequency of storytelling "events" and cooperation. For instance, they may be able to show that as skilled storytellers move from one camp to the other, cooperation levels increase.

An alternative strategy that the authors may follow to substantiate their causality claims is through priming. In the treatment condition, participants may attend some storytelling session by skilled storytellers before taking part in the cooperation game. In the control condition, such storytelling session would be absent (or even better, substituted by some individual activity, such as reading an essay or listening to a speaker in isolation from other people). The expected effect is that cooperation levels should be higher in the treatment condition.

We agree that these are both potential methods for establishing causality between storytellers and cooperation. However, given the level of investment (in time, money and resources) necessary to conduct fieldwork these are currently not possible to explore in the present study. We have included these in our discussion as potential methods to further explore the role of storytellers and act as a springboard for future research.

202-210: One potential option [to further test our hypothesis] would be to conduct longitudinal work to explore if patterns of camp-level cooperation vary with changes in camp composition, explicitly regarding the addition or loss of skilled storytellers. Logistically, however, this approach is very demanding, therefore a simpler approach to establish causality may be to use 'priming' experiments, such that those primed with a cooperative story ought to cooperate more in a subsequent task. Similar studies have previously been conducted (although not explicitly regarding storytelling), suggesting that individuals in experimental games are more cooperative if they can communicate and therefore coordinate their behaviour^{14,43}, consistent with the mechanism proposed here.

♣ Is the manuscript clearly written? If not, how could it be made more clear or accessible to nonspecialists?

The manuscript is written with accurate language and is broadly accessible. I would advise the authors to make all the necessary supporting and descriptive material accessible in this same paper. A reader cannot be expected to read a previously published paper by the authors to obtain relevant information.

Based on the reviewer's suggestions we have included additional details regarding experimental game design in the supplementary information. We clarify this in lines 352-355: 'We provide only a brief outline of the methodology here (for additional details about study design, such as pilot studies, the use of rice as a medium of exchange, designing the game structure and the relation of this game to other similar economic games, see the Supplementary Information).'

♣ Is the statistical analysis of the data sound, and does it conform to the journal's guidelines?

The analysis is sound. However the authors should specify the number of observations in each regression / test. They should also explain what a z-score is. Why are there 6,534 dyads?

In the Methods section we have elaborated on the definition and construction of z-scores (lines 273-274) and ensured that the number of observations is clear for each analysis. There are a large number of dyads (6,534 in the case of the camp-mate network) as each individual could select to live with any of their camp-mates, with each unique nominator-recipient pair as a separate dyad. Therefore, in a camp with 11 individuals, there would be 100 unique dyadic relationships (as self-nominations are impossible).

Below some other requests for clarification.

264-266: To allow all prestige measures across all individuals to be compared in the same model, all women were assigned a '0' for hunting and fishing, while all men were given a '0' for tuber-gathering.

This seems questionable.

We have created a new 'overall foraging skill' metric which controls for these sex differences:

322-330: Although including sex as a co-variate in analyses should control for these sex-specific foraging domains, to remove this potential confound we constructed an additional 'overall foraging skill' variable combining both male and female foraging activities (using average number of hunting and fishing nominations for male foraging skill, tuber-gathering nominations for female foraging skill, then merging the two). Using this new variable we find no qualitative difference in our main findings regarding who individuals chose as camp-mates or as recipients of cooperation (Supplementary Table 12), with storytellers still selected more frequently than skilled foragers.

514-515: Table 2: Models assessing the likelihood of skilled storytellers being selected in a 'camp-mate' network, using a logistic GEE regression (n = 291, dyads = 6,534).

How extensive is the network considered? Would for instance people from one municipality be allowed to evaluate links with people from the other municipality?

As described in lines 391-397, all data are within-camp; 'Once nominations for [...] non-camp-mates [...] were removed [...]. Between-camp dyads were coded as missing'. We have also updated the main text for clarity (lines 129-131) 'We asked 291 Agta across the 18 camps to choose who they would most like to live with (with a maximum of five nominations), obtaining 857 nominations out of a possible 6,534 dyads (all of which were within-camp nominations).'

572: Extended Data Table 3

Why proximity has a negative sign?

This is because higher 'proximity' values are for more distant individuals. A negative association means that more distant individuals were less likely to be chosen as future camp-mates (or less likely to receive resources in the game network). For clarity and ease of interpretation, 'proximity' has been now been renamed 'distance'.

References cited in authors' response:

Harari, Y. N. (2014). *Sapiens: A brief history of humankind*. Random House.

Purzycki, B. G., Apicella, C., Atkinson, Q. D., Cohen, E., McNamara, R. A., Willard, A. K., ... & Henrich, J. (2016). Moralistic gods, supernatural punishment and the expansion of human sociality. *Nature*.

Marlowe, F. (1999). Showoffs or providers? The parenting effort of Hadza men. *Evolution and Human Behavior*, 20(6), 391-404.

Macfarlan, S. J., & Lyle, H. F. (2015). Multiple reputation domains and cooperative behaviour in two Latin American communities. *Phil. Trans. R. Soc. B*, 370(1683), 20150009.

West, S. A., Griffin, A. S., & Gardner, A. (2007). Evolutionary explanations for cooperation. *Current Biology*, 17(16), R661-R672.

Delton, A. W., Krasnow, M. M., Cosmides, L., & Tooby, J. (2011). Evolution of direct reciprocity under uncertainty can explain human generosity in one-shot encounters. *Proceedings of the National Academy of Sciences*, 108(32), 13335-13340.

Rand, D. G., Peysakhovich, A., Kraft-Todd, G. T., Newman, G. E., Wurzbacher, O., Nowak, M. A., & Greene, J. D. (2014). Social heuristics shape intuitive cooperation. *Nature communications*, 5.

Reviewer #2 (Remarks to the Author):

We thank the reviewer for their insightful comments and are pleased with their overall positive review. We have addressed all of the reviewer's major comments, including a content analysis of hunter-gatherer stories, issues regarding causality, the potentially mitigating impact of mobility and why camp-mates would cooperate with skilled storytellers, as well as other minor issues noted. We believe that we have replied to all the reviewer's comments, and that our analyses support the main arguments of our article, namely that hunter-gatherer stories appear designed to coordinate social and cooperative behaviour, that the presence of skilled storytellers is associated with increased cooperation, and that skilled storytellers receive fitness benefits, both in terms of enhanced social support and increased reproductive success. These findings are consistent with our hypothesis that stories evolved to coordinate social behaviour and promote cooperation among hunter-gatherers, as well as providing a mechanism by which group-beneficial behaviour (storytelling) can evolve via individual-level selection. We believe that these conclusions are novel, supported by rigorous analyses, and of sufficient interest to a wide range of researchers to merit publication in Nature Communications. We hope that these revisions address the issues raised and the reviewer agrees that our paper represents an important and relevant contribution to the study of human behaviour and will consider our manuscript for publication. The reviewer's comments are in black font with our replies in red font.

This paper should be of interest to all researchers concerned with the evolution of human cooperation. Many researchers have argued that language at least facilitates, and perhaps is even required, for the extraordinary degree of collective action (group cooperation) in our species. But very few have focused on narrative (storytelling) as done here, and I cannot think of any cases where the role of storytelling in enhancing cooperation has been subject to careful empirical test. The paper very clearly lays out the claims, the evidence, and the methods used to collect this evidence.

The paper makes three central claims: 1) storytelling promotes group cooperation; 2) storytellers gain fitness benefits; and (the broadest and most significant) 3) the "results provide a plausible pathway by which a group-beneficial behaviour such as storytelling might have evolved through individual-level selection".

The evidence presented for these claims can be summarized as follows (numbers matching those I list above for the claims):

- 1) Agta stories have content promoting norms of cooperation and egalitarianism. The presence of good storytellers (as measured by free-listing nominations) increases camp cooperation (as measured using a modified Dictator game).
- 2) Storytellers are preferred as co-residents (as measured by the correlation between storytelling reputation and free-listing nominations of preferred camp mates).

Storytellers have higher reproductive success on average.

3) Given above, there is no free-rider problem with storytelling that would require group-level selection.

There are some issues with the evidence (again, numbering corresponds to above):

1) The sample of stories is very small ($n=4$); however, the paper attempts to blunt this problem with reference to stories from other hunter-gatherer societies (see "Extended Data Table 1"). There is no formal analysis of content -- the claim that stories promote group cooperation is impressionistic. No information is provided on individual mobility between camps; if it is high (which is known to be the case in many but not all other hunter-gatherer societies), and includes storytellers, it would weaken the case for seeing the correlation between presence of good storytellers and increased camp cooperativeness. Finally, the Dictator game is not on the face of it a good way to measure willingness to engage in collective action (group-level cooperation), as it provides the same score for giving $x\%$ to one other player as $x\%/n$ to n other players.

We will deal with each of these issues in turn. Firstly, following the suggestions of the reviewer we have conducted a literature search of stories found in seven forager societies and quantified the frequency of different story types (coding for social, cosmological, resource use, and natural phenomena content). As we now describe in lines 93-97: 'To explore the content of hunter-gatherer stories in greater detail we collected 89 stories over seven different forager societies and coded them according to subject matter (see Methods for further details). Of these stories, ~70% were classified as pertaining to 'social behaviour' (i.e., prescribing social norms or coordinating behavioural expectations), more than any other category (see Supplementary Table 2).' This bolsters our hypothesis that storytelling may have evolved to coordinate social behaviour and facilitate cooperation.

For additional details on this analysis see lines 480-499: 'In addition to the stories concerning social behaviour (Supplementary Table 1), we also performed a content analysis of all stories collected ($n=89$) from these Southeast Asian hunter-gatherer populations in addition to two African forager groups (Ju/'hoansi and BaYaka; Supplementary Table 2). Each story was assessed for different types of content: social (content which prescribed behaviour during interactions with others, such as cooperation, sex equality, norm-breaking, sex roles, punishment and interactions with out-groups); cosmological (content concerning the origins of the earth/universe); natural phenomena (content concerning navigation, animal origins/behaviour, fire and natural disasters/weather); and resource use (content about foraging and resource extraction). Each of these broad themes are common in forager folklore^{15,23,27,41,61}. Many stories were classified along more than one criterion, such as many creation stories combining cosmological and social content (see Supplementary Table 1). Some sources presented several versions of the same basic story; to prevent unnecessary duplication, in these cases only the longest or most elaborate stories were used in this analysis. Any analysis

of this kind will of course be subject to bias to some extent, such that others may categorise stories differently or their 'emic' interpretation in the specific society may be different. However, as social content was present in ~70% of stories, approximately double that of the second-highest content (natural phenomena; 36%), it is unlikely that any small changes in categorisation would greatly influence our conclusion that stories in hunter-gatherer societies appear to function to coordinate social behaviour.'

Secondly, we agree that mobility may be a mitigating factor regarding the association between storytellers and cooperation. We have collected indices of camp stability (the level of turnover in camp composition over multiple visits to the same camp) for a subset of 11 (out of 18) camps and can include these in our analysis to control for this effect. When including camp stability as a covariate in our analysis, we show that camps with a greater proportion of skilled storytellers are still associated with increased cooperation (Supplementary Table 4), suggesting that the effects of storytellers on cooperation are relatively independent from those of repeated interactions/mobility. Regarding the mobility of storytellers between camps, we have also included a discussion about how future studies can use the mobility of storytellers to further test and establish causality between storytelling and increased cooperation (i.e., testing whether if a skilled storyteller moves to a new camp cooperation subsequently increases):

199-205: The evidence provided here is consistent with the theory that storytelling acts as a mechanism to coordinate group behaviour and promote cooperation. However, these findings are largely correlational and further studies are required to conclusively demonstrate that storytelling performs a causal role in facilitating cooperative behaviour. One potential option would be to conduct longitudinal work to explore if patterns of camp-level cooperation vary with changes in camp composition, particularly regarding the addition or loss of skilled storytellers.

Thirdly, while multiple donations to the same individuals are possible, in practice only one individual (out of 290) gave more than one token to the same individual, so this is unlikely to greatly influence our argument regarding camp-level cooperation.

2) The link between storytelling reputation and camp-mate preference is purely correlational, making it difficult to rule out alternative explanations (as is also the case with the evidence for claim 1). However, the researchers did control for many other variables that plausibly could drive this correlation, which increases our confidence that the claim is supported.

We do acknowledge that our analyses are largely correlational, and have clarified this in the text where necessary. Specifically, we now state that:

30-31: [...] the presence of good storytellers *is associated with* increased levels of cooperation.

35-36: [...] storytelling *may be* an important mechanism to organise cooperative systems in hunter-gatherer societies.

111-113: [...] a 1% increase in nominations of good storytellers *was associated with an* increase in donations by 2.2 percentage points.

179-180: We conclude that storytelling *may be* an adaptation to organise cooperative systems in hunter-gatherer societies [...].

220-221: In sum, we argue that storytelling *may be* an adaptation to organise cooperation in hunter-gatherers [...].

224-226: From simple storytelling to complex religion and later formal institutions such as nation states, the evolution of storytelling *may have* been pivotal in organising and promoting human cooperation.

We have also included a new paragraph which specifically addresses these concerns regarding causality (lines 199-202): 'The evidence provided here is consistent with the theory that storytelling acts as a mechanism to coordinate group behaviour and promote cooperation. However, these findings are largely correlational and further studies are required to conclusively demonstrate that storytelling performs a causal role in facilitating cooperative behaviour.'

3) The evidence that storytellers have higher average RS does address the free-rider problem with regard to why be a storyteller rather than a passive listener. However, it does not answer the question of why others should provide hypothesized "social support" to storytellers (at presumably some personal cost), so the claim that individual-level benefits account for the system is incomplete.

This is a valid point and we thank the reviewer for highlighting this and we acknowledge that we do not explicitly answer why camp-mates should preferentially cooperate with storytellers. One potential reason for this may be that individuals cooperate with skilled storytellers in what may be trade-like relationships; individuals cooperate with storytellers so that they will tell stories and promote cooperation, although this requires additional research to explore in greater detail.

Indeed, we have included a new analysis in our revised manuscript (Supplementary Tables 10 & 11) which demonstrates that skilled storytellers are also more likely to receive resources than non-skilled storytellers in the experimental game (in addition to being preferred camp-mates). This suggests a potential pathway by which listeners who cooperate with storytellers may increase their fitness. While acknowledging that this pathway is speculative, we have therefore amended the text (line 170-176) to read: '[...] we also demonstrate that skilled storytellers are more likely to be recipients of resource transfers in the experimental game (Supplementary Tables 10 & 11). This suggests that

storytellers may be 'rewarded' for their public good by other camp-mates who benefit from the increased cooperation which storytellers may promote, in what may be mutually-beneficial trade-like relationships (although the individual-level benefits to those who cooperate with storytellers remain in need of further study).'

A minor quibble: Refs 20 & 21 are too general to support the claim that storytelling "is a costly behaviour requiring an input of time and energy into practice, performance and cognitive processing"; it could well be that some individuals just have "the gift" and it comes easily (and of course brings social rewards). However, this is not critical for evaluating the key claims listed above.

We agree, and have used these (admittedly rather general) references to support the claim that storytelling is a costly behaviour as little evidence assessing the actual costs of storytelling exists in the literature. Although impressionistic, but supporting this claim, several ethnographic sources highlight the theatrical and active aspect of storytelling, which would also increase the energetic costs of storytelling. We have amended this section to now read (lines 152-154): '[...] time and energy into practice, performance and cognitive processing^{33,34}. Indeed, several ethnographic sources highlight the theatrical and active nature often associated with storytelling performances^{25,27}.'

In sum, this is an impressive piece of research. Although I am not qualified to judge the statistical analyses in detail, the methods are laid out clearly, and seem careful enough, that I have no technical concerns about the analyses. Despite some issues with the degree to which the evidence adequately tests the claims (described above), I do think the paper is worthy of publication, given some revisions.

Reviewer #3 (Remarks to the Author):

We thank the reviewer for their insightful and detailed comments, which we have addressed and included as fully as possible in our revised manuscript. In particular, we have addressed the criticism that our comparative dataset of stories was previously too geographically-restricted to assess the importance of storytelling for hunter-gatherer cooperation. A new content analysis of 89 stories from seven different hunter-gatherer societies (including societies from Africa as well as Southeast Asia) has now been conducted, demonstrating that nearly three-quarters of all stories contain content which broadcasts social and cooperative norms, consistent with our hypothesis. We have also updated our manuscript in response to the reviewer's claim that we do not provide conclusive evidence that storytelling exhibits design to solve an adaptive problem by including a new introductory paragraph detailing the adaptive problem (cooperation) which we propose storytelling may help solve. Other issues raised by the reviewer have also been addressed, including issues of causality, other functional properties of stories, the inclusion of additional relevant literature, and clarification over other minor comments. We believe that we have replied to all the reviewer's comments, and that our analyses support the main arguments of our article, namely that hunter-gatherer stories appear designed to coordinate social and cooperative behaviour (among other functions), that the presence of skilled storytellers is associated with increased cooperation, and that skilled storytellers receive fitness benefits, both in terms of enhanced social support and increased reproductive success. These findings are consistent with our hypothesis that stories evolved to coordinate social behaviour and promote cooperation among hunter-gatherers, as well as providing a mechanism by which group-beneficial behaviour (storytelling) can evolve via individual-level selection. We believe that these conclusions are novel, supported by rigorous analyses, and of sufficient interest to a wide range of researchers to merit publication in Nature Communications. We hope that these revisions address the issues raised and the reviewer agrees that our paper represents an important and relevant contribution to the study of human behaviour and will consider our manuscript for publication. The reviewer's comments are in black font with our replies in red font.

Summary of Article

The major claims of this paper are that storytelling and fiction are an adaptation whose function is to promote cooperation in hunter-gatherer societies. In support of this claim, they present evidence that: (1) the Agta prefer skilled storytellers as group members over other skill specialists such as accomplished fishermen and hunters; (2) good storytellers have higher reproductive success; (3) Agta stories communicate information relevant to cooperation and egalitarianism; and (4) the presence of good storytellers increases cooperation in camps.

Methods

The investigators collected four Agta stories that elders “normally tell children and each other” (63). Each of these stories communicated norms and principles regulating social behavior: one story conveyed information about sex equality; one conveyed information about social egalitarianism; one conveyed information about group cooperation; and one conveyed information about group identity. This information includes an emphasis on the benefits of cooperation, examples of punishment for violating behavioral norms, and references to the use of reverse dominance hierarchies to prevent monopolization of power. They note that these themes occur in other hunter-gatherer groups (Extended Data Table 1): their evidence consists of a total of 11 stories from 6 other Southeast Asian hunter-gatherer groups (including another Agta population).

As the authors note, the higher degree of cooperativeness (as measured by the Dictator Game) found in groups with more storytellers is consistent with skilled storytellers spreading cultural norms, but is not conclusive because other explanations for these results are possible. Similarly, the findings that storytellers have higher reproductive success and that group members prefer skilled storytellers do not directly speak to the question of whether storytelling/fiction is an adaptation. As for the study sample, the size (four stories from the Agta) is adequate: the fact that these themes occur in stories that are popular in Agta culture is unequivocal evidence that people tell stories about these subjects. However, because they are limited to the same geographical region, neither the Agta data nor the comparative data provides compelling evidence for the larger claim that storytelling is an adaptation and that it evolved to promote cooperation.

We will respond to each of the issues mentioned by the reviewer in turn. Firstly, we acknowledge that the association between the proportion of skilled storytellers and increase in camp-level cooperation is correlational and other explanations are possible. We have therefore included a new paragraph which specifically addresses these concerns regarding causality (lines 199-202): ‘The evidence provided here is consistent with the theory that storytelling acts as a mechanism to coordinate group behaviour and promote cooperation. However, these findings are largely correlational and further studies are required to conclusively demonstrate that storytelling performs a causal role in facilitating cooperative behaviour.’ Nonetheless, as detailed in lines 107-111 and expanded upon in the Methods section (lines 380-382) we included additional variables to control for effects of relatedness and group size, which have previously been found to influence cooperative behaviour, and still report a significant effect of storytellers on camp cooperation. Furthermore, we have included a new analysis (Supplementary Table 4) using ‘camp stability’ (a proxy for frequency of repeated interactions) as another co-variate and still find increased cooperation in camps with a greater proportion of skilled storytellers. This gives us confidence that these results are a consequence of storytelling, although future research to conclusively establish causality is required.

Secondly, we would disagree over the reviewer's claim that storytellers possessing greater reproductive success and being preferred social partners does not address whether storytelling is an adaptation. An adaptation is a phenotype which derives inclusive fitness benefits for an organism. Both of the findings mentioned (in addition to our new analyses showing that skilled storytellers are preferred recipients of cooperation in the experimental game; Supplementary Tables 10 & 11) demonstrate that it is fitness-enhancing – that is, it is adaptive – to be a skilled storyteller. As stated in our manuscript (lines 190-193) 'the value of good storytellers is reflected in the fact that they also have increased reproductive fitness and receive more resources than less-skilled storytellers. We therefore provide a pathway by which storytelling, a group-beneficial behaviour, can evolve via individual-level selection.'

Thirdly, while we did not include stories from other hunter-gatherer populations outside of Southeast Asia in our original publication, we did write that (lines 455-461) 'stories by the Ju/'hoansi from southern Africa^{23,27} and Central African pygmies²⁵ were also sought [...] among the Ju/'hoansi many stories concern marriage behaviour, in-law relationships, kinship networks and sharing norms²³, as well as sex equality and the different (but complementary) roles of each sex^{27,57}. Similar trends are found among central African pygmy groups, such as the BaYaka²⁵'. This suggests that similar themes are found over other hunter-gatherer populations beyond Southeast Asia. Nonetheless, we have now conducted a literature search of stories found in seven forager societies (five from southeast Asia and two from Africa) and quantified the frequency of different story types (coding for social, cosmological, resource use, and natural phenomena content). As we now describe in lines 93-97: 'To explore the content of hunter-gatherer stories in greater detail we collected 89 stories over seven different forager societies and coded them according to subject matter (see Methods for further details). Of these stories, ~70% were classified as pertaining to 'social behaviour' (i.e., prescribing social norms or coordinating behavioural expectations), more than any other category (see Supplementary Table 2).' This comparative data from various geographical regions bolsters our hypothesis that storytelling may have evolved to coordinate social behaviour and facilitate cooperation (this does not mean stories cannot have other functions as well, but these findings overwhelmingly support our central hypothesis).

For additional details on this analysis see lines 480-499: 'In addition to the stories concerning social behaviour (Supplementary Table 1), we also performed a content analysis of all stories collected ($n=89$) from these Southeast Asian hunter-gatherer populations in addition to two African forager groups (Ju/'hoansi and BaYaka; Supplementary Table 2). Each story was assessed for different types of content: social (content which prescribed behaviour during interactions with others, such as cooperation, sex equality, norm-breaking, sex roles, punishment and interactions with out-groups); cosmological (content concerning the origins of the earth/universe); natural phenomena (content concerning navigation, animal origins/behaviour, fire and natural disasters/weather); and resource use (content about foraging and resource extraction).

Each of these broad themes are common in forager folklore^{15,23,27,41,61}. Many stories were classified along more than one criterion, such as many creation stories combining cosmological and social content (see Supplementary Table 1). Some sources presented several versions of the same basic story; to prevent unnecessary duplication, in these cases only the longest or most elaborate stories were used in this analysis. Any analysis of this kind will of course be subject to bias to some extent, such that others may categorise stories differently or their 'emic' interpretation in the specific society may be different. However, as social content was present in ~70% of stories, approximately double that of the second-highest content (natural phenomena; 36%), it is unlikely that any small changes in categorisation would greatly influence our conclusion that stories in hunter-gatherer societies appear to function to coordinate social behaviour.'

Content

(23) "Storytelling occurs spontaneously in childhood"

The reference cited in support of this claim (Boyd 2009) is unsuitable. Boyd is a literary scholar, not a developmental psychologist, has no training in developmental psychology, and does not do quantitative research on the acquisition of narrative by children. Much more compelling evidence comes from the work of Pitcher & Prelinger (1963) and Sutton-Smith (1981), who collected large samples of stories generated by preschool and elementary school children. For a review of this research, see Scalise Sugiyama 2009.

Although we have now removed all references from the abstract (to correspond to journal guidelines), we included the line 'storytelling [...] occurs spontaneously in childhood¹⁷' (line 61) and have referenced Scalise Sugiyama (2009) here, which we agree is a more apt reference to support this claim.

(24) "has the power to increase empathy towards others"

This conclusion is premature and only weakly supported. The Djikic et al. (2009) study cited by the authors shows that a fictional story arouses more emotion than a "court document meant to represent an ostensible divorce proceeding," but does not show that reading literature increases empathy. Indeed, Djikic et al.'s experiment does not test directly for empathy. Moreover, subjects were administered the Emotions Checklist immediately after reading either the Chekov story or the "court document," which provides no information about any hypothesized long-term effects of reading literature. Kidd & Castano (2013) similarly found that, compared to reading nonfiction, popular fiction, or nothing at all, reading literary fiction temporarily enhances theory of mind, as measured using the Reading the Mind in the Eyes Test. Again, however, their study does not speak to the long-term effects of reading literature vis-à-vis ToM abilities. Furthermore, Panero et al. (2016) recently failed to replicate Kidd & Castano's (2013)

findings using three independent research groups and a total of 792 participants. Other studies have found that the effects of reading fiction are moderated by individual difference variables such as transportation into the story, affective empathy, and openness to experience (e.g., Bal & Veltkamp 2013, Johnson 2012, Djikic et al. 2013).

We thank the reviewer for their knowledge on this topic and the literature provided. In our discussion we also refer to this research and have updated our language and references (including the failed replication) accordingly (lines 187-190): 'By introducing individuals to situations beyond their everyday experience, narratives may also increase empathy and perspective-taking towards others, including strangers³⁹⁻⁴¹(although see⁴²), potentially facilitating camp-level coordination and cooperation'. As this point about increasing empathy is largely tangential to our argument, we have removed this phrase from the abstract. Additionally, although enhancing empathy/ToM may be an interesting (if not conclusively proven) consequence of stories, it is not central to our central premise, but rather a potential proximate pathway by which storytelling may act to promote cooperation.

(28-29) "Despite its undeniable importance, little attention has been given to understanding the function and evolution of human storytelling"

Obviously, this is a matter of opinion, but there is certainly more research on this subject than the authors indicate. See, for example Minc 1986; Scalise Sugiyama 1996, 2005, 2008, 2011, 2012, 2014; Sobel & Bettles 2000; Steen & Owens 2001; Tooby & Cosmides 2001.

It was not our intention to overlook this existing research (and in our original manuscript we did cite several sources which explore this topic), but compared to other research areas the evolution of storytelling does remain an under-explored topic. There is also a lack of empirical testing of these hypotheses using real-world data from hunter-gatherer populations. We have now updated our reference list to reflect this existing literature and included a new sentence in the introduction to read (lines 76-79): 'Although others have proposed that storytelling was an important step in human evolution^{15, 19-22}, no studies have empirically tested this hypothesis using real-world data. For these reasons, we decided to analyse the content and functions of storytelling in a hunter-gatherer population'.

(29-30) "Here we propose that fiction and storytelling promote cooperation in hunter-gatherer groups"

The hypothesis that storytelling is used to promote cooperation is not novel. See Minc 1986; Sobel & Bettles 2000; Scalise Sugiyama 2011. Also, fiction and storytelling are not the same thing (Tooby & Cosmides 2001; Scalise Sugiyama 2012). There is a large

literature in psychology on fiction (a.k.a., pretense), which is widely viewed as a manifestation of the capacity for reasoning counterfactually (a.k.a. conditional reasoning; Leslie 1987; Cosmides & Tooby 2000; Onishi et al. 2007). This, in turn, is the foundation of perspective taking--i.e., imagining the environment from different spatial, temporal, or psychological perspectives (see, e.g., Schacter et al. 2007), the latter of which is typically referred to as theory of mind.

We acknowledge that the theory of storytelling promoting cooperation is not new (although explicit empirical testing has not previously been conducted) and have now updated our manuscript and references to read (lines 58-60): '[...] here we propose that storytelling in particular may play an essential role in the evolution of human cooperation by broadcasting social and cooperative norms to coordinate group behaviour (see also¹⁵)'

While there are undoubted cognitive differences between fiction and storytelling, we did not mean to imply that fiction in general promotes cooperation, but rather that stories – including fictional stories – promote cooperation. For clarity, we have removed 'fiction' from this line, which now reads (lines 24-25): 'Here we propose that storytelling promotes cooperation in hunter-gatherer groups'.

(39-40) "We conclude that storytelling is an important adaptation to organise cooperative systems in hunter-gatherer societies"

Earlier in the paper, the authors include fiction in this claim—"fiction and storytelling promote cooperation in hunter-gatherer groups" (29-30). The claim needs to be clarified (are both fiction and storytelling adaptations, or only storytelling?), especially since fiction and storytelling are not the same thing (see previous comment). Additionally, in order to claim that storytelling is an adaptation, the authors need to define it cognitively (story grammar research is a good place to start—e.g., Rumelhart 1975; Schank 1975; Thorndyke 1977). Their provisional definition ("an account of a sequence of events in the order in which they occurred to make a point") is incomplete and cognitively nebulous. Finally, the authors must demonstrate that there is an adaptive problem to be solved, and that storytelling exhibits design aimed at solving this problem. The present article does not discuss design, nor does it demonstrate that storytelling performs a function essential to cooperation. Moreover, the article glosses over that fact that forager oral storytelling transmits other types of information besides social norms: "Not all stories obtained from these populations concerned social behaviour. Many stories not described here concern cosmological content, with seemingly little social relevance. Examples include Batek origins of the cosmos, Maniq origins of night and day (originating from a snake continuously eating and regurgitating its tail), and Andamanese origins of fire (see references in Extended Data Table 1). Their existence does not diminish the importance of stories in facilitating cooperation, but merely highlights that stories can perform other functions, such as disseminating fitness-relevant information" (385-391). As the authors

concede, storytelling transmits other types of adaptively relevant knowledge. For example, one common theme in forager oral tradition is dangerous animals; thus, to make the case that storytelling evolved to “organise cooperation,” one must demonstrate that storytelling did not evolve in response to other adaptive problems, such as predator avoidance. Humans cooperate all the time without storytelling, which belies the claim that storytelling evolved and is designed expressly to facilitate cooperation. For a discussion of storytelling vis-à-vis cooperation, see Scalise Sugiyama’s (2008:31-32) article about tricksters, as well as Scalise Sugiyama 2011 (12-14) on storytelling and the broadcasting of social norms.

There are several points to cover here, so we will take each in turn (we will not discuss storytelling and fiction again, as this was covered in the previous response). Firstly, regarding the need to define storytelling cognitively, we feel that our definition encompasses the main functions of a story (i.e., a plot in order to make a point). Nonetheless, we have also added a more specific definition of what constitutes a story based upon ‘story grammar research’, see lines 257-260: ‘A story can also be defined by its components, and several lines of evidence have converged from literary theory and cognitive psychology which suggest that narratives consist of: character, setting, events, causal connections and resolution (for a review see²²)’.

Secondly, the problem of cooperation is a central problem in biology, and is therefore a problem in need of an adaptive solution (there is a clear problem that needs to be solved, and it is a central debate in human evolution), especially given our species’ reliance on cooperation to survive. Perhaps our initial submission did not discuss this in enough detail, so we have now included a new introductory paragraph explaining why cooperation is an adaptive problem to be solved.

45-60: Cooperation is a central problem in biology^{1,2}. This is especially true in humans given the range of extensive cooperation observed, including food-sharing^{3,4}, allocare^{5,6} and political coalitions⁷. Adaptive explanations for cooperation often focus on the ‘free-rider problem’; that is, explaining how a behaviour which decreases fitness (at least in the short-term) can be evolutionarily advantageous. Many solutions to this problem have been proposed, such as kin selection⁸, reciprocal cooperation⁹, costly signalling¹⁰ and indirect reciprocity¹¹, among others. However, even in situations where cooperation would be the best strategy for all involved, cooperation may not occur due to ‘problems of coordination’. Under these circumstances cooperation is not hindered by the potential for free-riding, but rather a lack of common knowledge over the behaviour of others^{12,13}. Meta-knowledge is therefore required to solve these problems of coordination. In other words, it is not enough to know how to act in a given situation, individuals need to know that others also know how to act. While language is undoubtedly essential as a medium of communication for coordination¹⁴, here we propose that storytelling in particular may play an essential role in the evolution of human cooperation by broadcasting social and cooperative norms to coordinate group behaviour (see also¹⁵).

Thirdly, we would disagree with the reviewer's claim that the article does not 'demonstrate that storytelling performs a function essential to cooperation'. As we have shown in our article, a greater proportion of skilled storytellers in camp is associated with increased levels of cooperation, consistent with storytellers promoting cooperation in camps. Additionally, a great number of stories from hunter-gatherer populations concern cooperative and social behaviour (see new content analysis, above), further suggesting that much forager storytelling performs the function of coordinating such behaviour.

Finally, while we have focussed on the role of storytelling to promote cooperation, this hypothesis is not mutually exclusive from storytelling performing other functions, as we stated in our original submission. As the reviewer makes clear (and we do not disagree with), storytelling is a multi-faceted phenomenon which likely solves many other adaptive problems, such as predator avoidance, learning foraging skills, navigation, natural disaster preparation, among other potential functions (as Scalise Sugiyama (2012; page 364) states: 'Narrative is inherently polysemous' – We could not agree more). While we did not intend to give the impression that these functions were unimportant, our paper focuses primarily on its role in coordinating social behaviour and promoting cooperation. Indeed, in lines 194-198 we reference this research and state that: 'Although narratives are known to serve other adaptive functions, such as disseminating information on survival, foraging and the environment^{15,24,41,42}, or simply to entertain and hold the audience's attention²¹, we have shown that storytelling may have also been an important factor in the origins of widespread human cooperation'.

(70-73) "while also exemplifying various mechanisms of social norm enforcement, such as emphasising the benefits to cooperation over competition, examples of punishment for breaking norms, and reverse dominance hierarchies to prevent individual accumulation of power"

For a discussion of the use of storytelling for intentional leveling and social sanctioning, see Scalise Sugiyama 2011, 2012.

We have updated our references in this section to include this previous research (line 92).

(125-126) "what would be the individual benefit for storytellers"

This question has been addressed and answered elsewhere. See Scalise Sugiyama 2005, 2011, 2012.

We have updated our references in this section to include this previous research (line 149), although we do note that none of this previous research empirically tested whether storytellers gained fitness benefits. Additionally, two of these references (Scalise

Sugiyama 2011; Scalise Sugiyama & Sugiyama 2012) discuss the individual benefits of storytelling as accruing via parent-offspring transmission (i.e., intergenerational transmission via kin selected mechanisms). While we do not disagree with this putative benefit, our analyses suggest a separate fitness-enhancing pathway to increased fitness via increased social support and cooperation from other adult camp-mates.

(142-143) "It is possible that by performing an important social function, skilled storytellers receive increased social support from others"

This idea has been proposed elsewhere: see Scalise Sugiyama & Sugiyama 2003.

We have also included this reference in our revised manuscript (line 168).

Recommendation

Of value in this study is the evidence it provides from 7 different Southeast Asian hunter-gatherer groups that storytelling transmits information related to social norms. This finding needs to be integrated into the large body of research on the function (evolved or otherwise) that storytelling serves in preliterate, small-scale societies, most of which the authors do not cite. The same goes for the authors' claims regarding storytelling and empathy: their review of this literature is thin and problematic. Finally, the study does not provide convincing theoretical or quantitative evidence in support of its claim that storytelling/fiction is an adaptation or that its function is to "organize cooperation." For these reasons, I do not find the article suitable for publication in Nature Communications.

We again thank the reviewer for their useful comments and constructive feedback. We believe that our revised manuscript addresses each of the criticisms made by the reviewer against publication. Specifically, our new content analysis of 89 stories from seven different hunter-gatherer populations (from societies in Africa, as well as Southeast Asia) demonstrates that the majority of stories told by hunter-gatherers concern broadcasting social and cooperative norms. This content analysis, along with our findings indicating that camps with a greater proportion of storytellers display greater levels of cooperation, are consistent with our hypothesis that storytelling may have evolved to coordinate social behaviour and promote cooperation among hunter-gatherers. A new introductory paragraph has also been added which details the adaptive problem (cooperation) which we propose storytelling may help solve by coordinating social behaviour. As detailed above, our revised manuscript also contains several additional references to previous work exploring the evolution of storytelling among small-scale societies to further situate our study within the wider existing literature on this topic. We have also clarified our position on the association between storytelling and empathy, as well as the distinction between storytelling and fiction, in our revised manuscript and our responses above. We therefore believe that we have addressed each

of the reviewer's reasons against publication in Nature Communications, and hope that the reviewer agrees that our manuscript represents an important contribution to the field and will consider our paper for publication.

References cited in authors' response:

Scalise Sugiyama, M., & Sugiyama, L. S. (2012). Once the child is lost he dies': Monster stories vis-a-vis the problem of errant children. In *Creating consilience: Integrating the sciences and the humanities* (Eds. Slingerland, E & Collard, M), 351-371.

Scalise Sugiyama, M. (2011). The forager oral tradition and the evolution of prolonged juvenility. *Frontiers in psychology*, 2.

Reviewers' comments:

Reviewer #1 (Remarks to the Author):

I appreciate the additional work carried out by the authors and I find this resubmitted version substantially improved. Below are more detailed comments.

1) Causality

I find the more cautious language used by the authors fitting well with the correlational nature of their results. The extra paragraph added at lines 199-202 is particularly insightful. I would just add one sentence or two arguing that the two aspects object of the study, i.e. co-operation and story-telling, may have co-evolved creating a virtuous cycle of mutual reinforcement.

2) Measurement of storytelling abilities

I am happy with the additional analysis and the references provided by the authors. I am in no doubt that replacing the "relative" measure with an "absolute" measure of story-telling skills would have not considerably affected the results. Still, I think that the absolute measure is clearly superior to the relative one, and for sure it could help reducing unnecessary noise in the data. I would therefore encourage the authors to use this measure in their future work (I can't see how it would require greater time investment).

3) Definition of storytelling

I am satisfied with the changes introduced by the authors.

4) The experimental game measures altruism but not propensity to cooperate

The description of the game now introduced makes the design clearer. I agree that a non-anonymous game with the possibility of reciprocation fits the authors' goals, and I can see the reasoning behind defining such an interaction as "cooperative". Since this is a key aspect to qualify the research as properly addressing cooperation, I invite the authors to explicitly quote West et al (2007)'s definition, in order to clarify which operational measure of cooperation they have taken. It is obvious that there are different forms of cooperation, some of which involve mere coordination to achieve common goals, while in some others some parties have to sacrifice some fitness for the common good. Since influential work by Nowak (2006, cited by the authors) focus on the latter form of cooperation, I think the reader will benefit from this clarification.

I still believe that a Dictator Game of the type used by the authors is not optimal for their goals, even if the objective is to mirror food exchange in these societies, as the authors claim. Provided that food consumption has obvious decreasing returns (food will benefit infinitely a starving person while it will have no effect, or even harm, a satiated person), then a "helping game" in which the "recipient" receives a larger quantity of resources than the "sender" fits better the situation the authors want to capture. I think the discussion of

the literature proposing the idea that people mistake one-shot interactions for repeated interactions, which the authors use in their rebuttal letter, is superfluous in this context. Moreover, it confuses motivations – which may or may not be genuinely altruistic – with actions – which are in themselves fitness-decreasing or increasing regardless of the motivations behind. I think a sentence of the kind “This game was designed to be intuitive to grasp for the Agta without the need for complex explanations which may lead to frustration, boredom and misunderstandings, as well as to mimic food-sharing decisions made on a daily basis in this society” is unnecessary. It indirectly implies that in societies where more complex experimental games have been played subjects may have been frustrated, bored and not reaching full understanding, which is completely unwarranted. I wouldn’t even say that this game mimics food sharing, as just explained. Overall, I think the main conclusions of the paper will stand even with the current form of the DG, but I invite the authors to consider adopting Helping games for their future work.

On all other points, I am satisfied with the additional analyses provided by the authors.

Reviewer #2 (Remarks to the Author):

I am satisfied that the authors have thoroughly revised the manuscript and addressed most of the reviewers' comments and suggestions.

Reviewer #3 (Remarks to the Author):

This study presents some novel and exciting quantitative findings that support existing theoretical work (e.g., Sugiyama & Scalise Sugiyama 2003; Scalise Sugiyama 2011): (1) storytellers are highly valued by their fellow band members; (2) skilled storytellers have an additional .53 living offspring compared to non-skilled storytellers; and (3) “older individuals possessed a greater storytelling reputation” (295-296). However, the study’s central claim—that storytelling is an adaptation that evolved to promote cooperation—does not satisfactorily account for these results. Additionally, while the methods elicit important new information about the value of storytellers, they do not convincingly test the claim that storytelling is an adaptation for cooperation per se. The abovementioned findings merit publication, but not within the present theoretical framework (see discussion below). One way to get around this problem would be to modify the argument: instead of claiming that storytelling is an adaptation and that it is designed to promote cooperation, one could go with the more parsimonious claim that storytelling can provide fitness benefits in the realm of cooperation.

Storytelling as Adaptation

As I see it, the heart of the argument here is that, “if storytelling plays a functional role in promoting cooperation, we predict that skilled storytellers would be preferred as social partners. In contrast, if storytelling is only a consequence or by-product of cooperation, preferred social partners are likely to be chosen on the basis of other characteristics, such as foraging skills or medicinal knowledge” (119-123). I don’t understand the logic of this

conclusion, as we would expect a storyteller to be valued for “promoting cooperation” regardless of whether storytelling is an adaptation or merely a “by-product of cooperation.” Moreover, given that storytelling transmits several different types of useful information (as noted in the paper), this hypothesis begs the question: why did the benefit of transmitting information about social norms per se drive the evolution of storytelling? How do we know that storytellers aren’t preferred as social partners for their value as a font of useful knowledge of several kinds (e.g., animal behavior, topography, warfare)? To rigorously test the hypothesis that it was the transmission social norms—and not other kinds of information—that drove the evolution of storytelling, one must rule out the possibility that the use of storytelling to transmit other types of information confers the same benefits upon the storyteller.

Additionally, I did not see any statement of the theoretical reason for dismissing other hypotheses regarding the possible adaptive function or fitness benefits of storytelling—and there are several (e.g., Scalise Sugiyama 1996, 2017; Tooby & Cosmides 2001; Coe et al. 2006; Zunshine 2006; Mar & Oatley 2008; Boyd 2009). I do not understand why a review of the literature was not included in the paper. As it stands, the reader is unable to determine why alternative hypotheses have been rejected and why the cooperation hypothesis is more compelling than these alternatives. (Note: in the theoretical literature, storytelling [a.k.a. narrative, fiction, literature] is sometimes lumped with art behavior in general--e.g., Pinker 1994; Miller 2000; Hagen & Bryant 2003.)

In my previous review, I called for more attention to design. This is another area where a review of the literature might be helpful. A case in point is the possible benefits of storytelling to the storyteller and the audience—specifically, the claim that “storytellers may be ‘rewarded’ for their public good by other camp-mates who benefit from the increased cooperation which storytellers may promote, in what may be mutually-beneficial trade-like relationships (although the individual-level benefits to those who cooperate with storytellers remain in need of further study)” (172-174). While I agree that the potential benefits of storytelling need further study—especially quantitative study—there has been some theoretical work on the subject. Since the claim made in the paper is theoretical as well, I don’t understand why there is no reference to this previous work--e.g., Pinker 1994 (538-543); Scalise Sugiyama 1996 (411-412), 2005 (188-189); Tooby & Cosmides 2001. On this point, one of the most suggestive findings of the study is that the effect of age was “strongly significant” (308-309), that “older individuals possessed a greater storytelling reputation” (295-296), and that “many of the Agta rated by others as skilled storytellers were also those who were the most engaging and knowledgeable during our fieldwork” (347-348). These findings are consistent with the hypothesis that storytellers are valued because of the knowledge they possess (i.e., all of the knowledge transmitted via the stories, not just information about social norms). Humans continue to accumulate knowledge across the lifespan: thus, the longer a person has lived, the more knowledge he/she possesses. This may be the basis for the deep respect accorded to elders in indigenous oral cultures: although I don’t know of any study that has quantified this relationship, in the ethnographic literature elders are commonly characterized—both emically and etically--as the “libraries” of their society. On this point, it would be useful to know what the participants understood “skilled storyteller” to mean. A person who can tell a story in an engaging manner? A person who knows a lot of stories? A person who knows the stories in their entirety and can recount them without error? All of the above?

Finally, da Silva & Tehrani (2016) trace the origin of some Indo-European tales back about 6000 years. While this is an exciting study, 6000 years is not a “deep evolutionary history” (63), nor is it sufficient time for a complex cognitive adaptation to evolve. I support the claim that storytelling is ancient, but would recommend citing more compelling evidence of its antiquity.

Methods

I completely agree that “it is difficult to obtain an unbiased sample of stories told by hunter-gatherers as different ethnographers tend to focus on different story contents” (451-453). Given that this is the case, then, it is unclear what—if anything—frequency measures of content can tell us about the relative importance of the different types of information contained in collections of forager oral tradition. Moreover, it seems incautious to conclude that, because one type of information occurs more frequently than another, the former references a selection pressure while the latter does not. This appears to be what the paper is claiming, as evinced by the statement that, “as social content was present in ~70% of stories, approximately double that of the second-highest content (natural phenomena; 36%), it is unlikely that any small changes in categorization would greatly influence our conclusion that stories in hunter-gatherer societies appear to function to coordinate social behavior” (495-499). This seems tantamount to arguing that, because hunter-gatherer women spend more time gathering than giving birth, gathering exerted selection pressure on our female ancestors but parturition did not. To put it another way, because virtually all stories feature characters (i.e., agents), narrative processing involves making inferences regarding the characters’ mental states (e.g., beliefs, desires, feelings); thus, a frequency-based approach might lead one to argue, à la Zunshine (2006) and Mar & Oatley (2008), that storytelling organizes theory of mind rather than cooperation.

I am also curious as to how the story content analysis was conducted—especially, how information about animal behavior was operationalized. In forager oral tradition, much information about animal behavior is encoded in origin myths and etiological tales. For example, in a Koyukon story about the origin of the potlatch and the games associated with it, the animal-people have a race. The wolverine is in the lead, but then “each time he saw a little spruce tree he went to rub his butt on it He would have won but he never passed a spruce tree (without stopping to rub his butt on it) So I don’t know who won, maybe it was the wolf, it’s said that the wolf also travels very fast” (Attila 1983:55). These lines describe the wolverine’s habit of marking its territory using anal scent glands, and also note that both wolverines and wolves can travel rapidly through snow (the story is set in winter). Due to the anthropomorphic aspects of the characters, a naïve coder might miss this information, or mistakenly assume it was fanciful. So my concern is that the content analysis may underestimate the frequency with which information about animal behavior/traits occurs in forager oral tradition. Information about natural disasters, environmental hazards, subsistence stress, and wayfinding is similarly encoded and, consequently, may also be underrepresented in the content analysis.

As noted above, the finding that “storytelling skill is associated with increased fitness” is exciting, but it doesn’t speak to precisely what is valued in the skilled storyteller. Is it his/her ability to entertain others, promote cooperation, provide useful information, or something else? I find it impossible to conclude from these results that storytelling is an adaptation for cooperation per se.

Finally, to demonstrate the presence of design, a phenomenon must be shown to be species-typical. While a survey of 7 cultures is better than a survey of 1 (and a survey of 89 stories is better than a survey of 4), it is still inadequate to this task (and there is still the problem of 5 cultures in the sample being from the same geographic region). I would recommend using another, less labor-intensive approach, although I do not have any good suggestions at the moment.

References

- Attla, C. 1983. *As my grandfather told it: Traditional stories from the Koyukuk told by Catherine Attla*. Yukon-Koyukuk School District and Alaska Native Language Center.
- Boyd, B. (2009). *On the origin of stories*. Harvard University Press.
- Coe, K., Aiken, N. E., & Palmer, C. T. (2006, March). Once upon a time: Ancestors and the evolutionary significance of stories. In *Anthropological Forum* (Vol. 16, No. 1, pp. 21-40). Routledge.
- Hagen, E. H., & Bryant, G. A. (2003). Music and dance as a coalition signaling system. *Human nature*, 14(1), 21-51.
- Mar, R. A., & Oatley, K. (2008). The function of fiction is the abstraction and simulation of social experience. *Perspectives on psychological science*, 3(3), 173-192.
- Miller, G. (2000). *The mating mind: How sexual selection shaped the evolution of human nature*.
- Pinker, S. (1994). *How the mind works*.
- Scalise Sugiyama, M. (1996). On the origins of narrative. *Human Nature*, 7(4), 403-425.
- Scalise Sugiyama, M. (2017). Narrative. *Encyclopedia of Evolutionary Psychological Science*. Springer.
https://link.springer.com/referenceworkentry/10.1007/978-3-319-16999-6_3316-1
- Tooby, J., & Cosmides, L. (2001). Does beauty build adapted minds? Toward an evolutionary theory of aesthetics, fiction, and the arts. *SubStance*, 30(1), 6-27.
- Zunshine, L. (2006). *Why we read fiction: Theory of mind and the novel*. Ohio State University Press.

Reviewer #1 (Remarks to the Author):

We again thank the reviewer for their original insightful and detailed comments which greatly improved the paper, and are glad that they find the revised manuscript substantially improved. We discuss any additional comments by the reviewer below.

I appreciate the additional work carried out by the authors and I find this resubmitted version substantially improved. Below are more detailed comments.

1) Causality

I find the more cautious language used by the authors fitting well with the correlational nature of their results. The extra paragraph added at lines 199-202 is particularly insightful. I would just add one sentence or two arguing that the two aspects object of the study, i.e. co-operation and story-telling, may have co-evolved creating a virtuous cycle of mutual reinforcement.

We agree that the language of the paper has now improved and have added a new sentence detailing the potential co-evolution of cooperation and storytelling (lines 210-211): 'It is therefore possible that cooperation and storytelling co-evolved via a process of mutual reinforcement.'

2) Measurement of storytelling abilities

I am happy with the additional analysis and the references provided by the authors. I am in no doubt that replacing the "relative" measure with an "absolute" measure of story-telling skills would have not considerably affected the results. Still, I think that the absolute measure is clearly superior to the relative one, and for sure it could help reducing unnecessary noise in the data. I would therefore encourage the authors to use this measure in their future work (I can't see how it would require greater time investment).

We are glad that our new analyses convinced the reviewer of the robustness of our results and will certainly consider alternative 'absolute' measures of skill for our future work.

3) Definition of storytelling

I am satisfied with the changes introduced by the authors.

4) The experimental game measures altruism but not propensity to cooperate

The description of the game now introduced makes the design clearer. I agree that a non-anonymous game with the possibility of reciprocation fits the authors' goals, and I can see the reasoning behind defining such an interaction as "cooperative". Since this is a key aspect to qualify the research as properly addressing cooperation, I invite the authors to explicitly quote West et al (2007)'s definition, in order to clarify which operational measure of cooperation they have taken. It is obvious that there are different forms of cooperation, some of which involve mere coordination to achieve common goals, while in some others some parties have to sacrifice some fitness for the common good.

Since influential work by Nowak (2006, cited by the authors) focus on the latter form of cooperation, I think the reader will benefit from this clarification.

As the reviewer suggests, for clarity we have now included West et al. (2007)'s definition of 'cooperation' (although for brevity we have paraphrased their definition, rather than quoted verbatim: lines 46-47): 'Adaptive explanations for cooperation – broadly defined as a behaviour which evolved to benefit others...'. We also agree that there is often confusion in the cooperation literature regarding how to define such behaviour, and therefore chose as inclusive a definition as possible which includes acts which both enhance and harm an individual's direct fitness (mutual benefit and altruism).

I still believe that a Dictator Game of the type used by the authors is not optimal for their goals, even if the objective is to mirror food exchange in these societies, as the authors claim. Provided that food consumption has obvious decreasing returns (food will benefit infinitely a starving person while it will have no effect, or even harm, a satiated person), then a "helping game" in which the "recipient" receives a larger quantity of resources than the "sender" fits better the situation the authors want to capture. I think the discussion of the literature proposing the idea that people mistake one-shot interactions for repeated interactions, which the authors use in their rebuttal letter, is superfluous in this context. Moreover, it confuses motivations – which may or may not be genuinely altruistic – with actions – which are in themselves fitness-decreasing or increasing regardless of the motivations behind. I think a sentence of the kind "This game was designed to be intuitive to grasp for the Agta without the need for complex explanations which may lead to frustration, boredom and misunderstandings, as well as to mimic food-sharing decisions made on a daily basis in this society" is unnecessary. It indirectly implies that in societies where more complex experimental games have been played subjects may have been frustrated, bored and not reaching full understanding, which is completely unwarranted. I wouldn't even say that this game mimics food sharing, as just explained.

Overall, I think the main conclusions of the paper will stand even with the current form of the DG, but I invite the authors to consider adopting Helping games for their future work.

It was not our intention to imply that in other societies where more complex games have been conducted the participants were frustrated, bored and misunderstood the game. We have therefore removed this sentence.

As the reviewer suggests, an alternative game where the recipient receives a larger quantity than the giver donates could also be used to explore collective action and we will consider such games for our future work.

On all other points, I am satisfied with the additional analyses provided by the authors.

Reviewer #2 (Remarks to the Author):

I am satisfied that the authors have thoroughly revised the manuscript and addressed most of the reviewers' comments and suggestions.

We again thank the reviewer for their original insightful and detailed comments which greatly improved the manuscript, and are glad that they find the revised paper substantially improved.

Reviewer #3 (Remarks to the Author):

This study presents some novel and exciting quantitative findings that support existing theoretical work (e.g., Sugiyama & Scalise Sugiyama 2003; Scalise Sugiyama 2011): (1) storytellers are highly valued by their fellow band members; (2) skilled storytellers have an additional .53 living offspring compared to non-skilled storytellers; and (3) “older individuals possessed a greater storytelling reputation” (295-296). However, the study’s central claim—that storytelling is an adaptation that evolved to promote cooperation—does not satisfactorily account for these results. Additionally, while the methods elicit important new information about the value of storytellers, they do not convincingly test the claim that storytelling is an adaptation for cooperation per se. The abovementioned findings merit publication, but not within the present theoretical framework (see discussion below). One way to get around this problem would be to modify the argument: instead of claiming that storytelling is an adaptation and that it is designed to promote cooperation, one could go with the more parsimonious claim that storytelling can provide fitness benefits in the realm of cooperation.

We thank the reviewer for their helpful comments which greatly enhanced the manuscript, and are glad that they now agree that these findings merit publication. We appreciate the further detailed comments provided by the reviewer and hope that our responses below address any remaining concerns.

First of all, we would like to make clear that at no point do we argue that the *only* adaptive function of storytelling is to promote cooperation. We do not reject alternative functional interpretations of storytelling, such as disseminating information regarding animal behaviour, geography or survival, or as a form of ‘mental simulation’ (see quotations below). Our argument is not that ‘storytelling is an adaptation and that it is designed to promote cooperation’, but rather that *one of the adaptive functions of storytelling is to promote cooperation*. We believe that this clarification (addressed in more detail below and in changes to the manuscript) is in line with the reviewer’s suggestion that ‘one could go with the more parsimonious claim that storytelling can provide fitness benefits in the realm of cooperation’.

In addition to the three results mentioned by the reviewer, we would like to emphasise three other major findings presented by our study which highlight the importance of storytelling as a mechanism to coordinate social behaviour and promote cooperation among foragers: 1) Each of the four Agta stories we collected concerns social and cooperative behaviour; 2) In a content analysis of 89 stories from seven hunter-gatherer societies, ~70% of stories reflected social behaviour (more than any other category of story); and 3) Agta camps with a greater proportion of skilled storytellers were associated with increased levels of cooperation. It is these additional findings which lead us to conclude that one of the main functions of stories is to promote cooperation. We have also modified the text to clarify that our argument is that storytelling can promote cooperation, but may also perform other functions:

Lines 153-157: These results demonstrate that the Agta prefer to live in camps with skilled storytellers, who are even more valued than good foragers, which may reflect the importance of storytellers in promoting cooperation and bringing

gains to all individuals in a camp (although storytellers may also be favoured for disseminating other fitness-relevant information, such as foraging, survival and geography).

Lines 186-189: Alternatively, people might enjoy listening to stories for other reasons, such as a form of 'mental simulation' to learn about their social and physical environment, and be paying for this service (this effect could be independent from the function of storytelling in promoting cooperation).

Lines 206-210: Although narratives are known to serve other adaptive functions, such as disseminating information on survival, foraging and the environment, or simply to entertain and hold the audience's attention, we have shown that storytelling may also have been an important factor in the origins of widespread human cooperation.

Storytelling as Adaptation

As I see it, the heart of the argument here is that, "if storytelling plays a functional role in promoting cooperation, we predict that skilled storytellers would be preferred as social partners. In contrast, if storytelling is only a consequence or by-product of cooperation, preferred social partners are likely to be chosen on the basis of other characteristics, such as foraging skills or medicinal knowledge" (119-123). I don't understand the logic of this conclusion, as we would expect a storyteller to be valued for "promoting cooperation" regardless of whether storytelling is an adaptation or merely a "by-product of cooperation."

Our results suggest that storytellers are valued as camp-mates and are present in larger numbers in cooperative camps. We suggest that this is consistent with storytellers performing an adaptive function of promoting cooperation. However we acknowledge they could also be a consequence of higher levels of cooperation in these camps. As we state in the manuscript (lines 121-125): 'this association may have alternative explanations and result from other social processes. For example, more cooperative camps may tell a greater number of stories, perhaps because they are more socially cohesive'.

We however do not follow the logic of the reviewer's argument: How could storytelling be valued for "promoting cooperation" (cause) if it is only a by-product of cooperation (consequence), and not a "promoter of cooperation" (cause)? In our view, if storytelling was a by-product of cooperation, the reason for people to prefer storytellers as social partners should be other than the fact that it "promotes cooperation". This would also be a plausible explanation. However, the content analyses of the stories shows that a high proportion of them have cooperative contents, corroborating our conclusion (see below).

Moreover, given that storytelling transmits several different types of useful information (as noted in the paper), this hypothesis begs the question: why did the benefit of transmitting information about social norms per se drive the evolution of storytelling? How do we know that storytellers aren't preferred as social partners for their value as a font of useful knowledge of several kinds (e.g., animal behavior, topography, warfare)?

We believe the reviewer is asking: were storytellers preferred camp-mates – or achieved higher fitness – because they promoted cooperation, or for other reasons, such as transmitting other fitness-relevant information?

As stated above and in our manuscript, we do not claim that the only adaptive function of storytelling is to disseminate social information and promote cooperation. It is therefore plausible that storytellers are favoured for reasons beyond simply their ability to promote cooperation. However, we presented three results suggesting that storytellers are preferred social partners for promoting cooperation.

First, given that the majority of stories among the Agta concern social behaviour, this leads us to suggest that this may be one of storytelling's main functions and therefore one of the reasons camp-mates would prefer to live with skilled storytellers (in addition to why they would have increased reproductive success) Second, the analysis of 89 stories from a larger sample of seven forager groups corroborates this finding. Third, the association between cooperation and the proportion of storytellers in camp also suggests that this may be one of the main functions of stories, and therefore individuals would prefer to live with, cooperate with, and increase the fitness of, others who can encourage cooperation.

However, as stated in the text, our suggestion is compatible with storytelling serving other functions, as demonstrated by the fact that about 30% of stories refer to other forms of knowledge.

Ultimately, while our research does suggest that coordinating social behaviour may be one of the functions of stories – and therefore a plausible reason why camp-mates would prefer to live with storytellers and why storytellers have greater fitness – additional research is required to tease apart the precise reasons for these findings. For clarity, we have now updated our manuscript to read (lines 153-157): 'These results demonstrate that the Agta prefer to live in camps with skilled storytellers, who are even more valued than good foragers, which may reflect the importance of storytellers in promoting cooperation and bringing gains to all individuals in a camp (although storytellers may also be favoured for disseminating other fitness-relevant information, such as foraging, survival and geography)'. Nonetheless, even if storytellers are chosen as camp-mates or have increased fitness for other reasons in addition to their ability to promote cooperation, this does not significantly alter our main argument. Rather, it demonstrates the multi-faceted nature of storytelling (of which promoting cooperation may be an important part).

To rigorously test the hypothesis that it was the transmission social norms—and not other kinds of information--that drove the evolution of storytelling, one must rule out the possibility that the use of storytelling to transmit other types of information confers the same benefits upon the storyteller.

As stated above, we agree with the reviewer's comment that storytelling performs multiple functions in addition to promoting cooperation. However, we believe that demonstrating that transmission of cooperative norms contributed to the evolution of storytelling does not require the rejection of all other possible or current uses of storytelling. We believe that complex adaptations are likely to

result from multiple selective forces. For example, there are various theories for brain size evolution in primates, such as visual specialisation (Barton), social intelligence (Dunbar), and cultural intelligence (Laland, Tomasello, van Schaik). Although each proposes an original or main driver for brain size evolution, they all agree that a large brain confers multiple adaptive advantages in foraging, social coordination, cooperation, cultural transmission, and mate choice, among others.

We believe the same is true for storytelling. Our study attempted to identify the possible uses and adaptive value of storytelling. We found that storytelling correlates with levels of cooperation, and that most of the story content in hunter-gatherers is related to social norms and cooperation. However, this does not deny that storytelling serves other adaptive uses such as transmitting other forms of knowledge.

Additionally, I did not see any statement of the theoretical reason for dismissing other hypotheses regarding the possible adaptive function or fitness benefits of storytelling—and there are several (e.g., Scalise Sugiyama 1996, 2017; Tooby & Cosmides 2001; Coe et al. 2006; Zunshine 2006; Mar & Oatley 2008; Boyd 2009). I do not understand why a review of the literature was not included in the paper. As it stands, the reader is unable to determine why alternative hypotheses have been rejected and why the cooperation hypothesis is more compelling than these alternatives. (Note: in the theoretical literature, storytelling [a.k.a. narrative, fiction, literature] is sometimes lumped with art behavior in general--e.g., Pinker 1994; Miller 2000; Hagen & Bryant 2003.)

We believe it is an unfair criticism to require us to refute all hypotheses proposed by the authors listed by the reviewer. The main objective of our manuscript is to provide a test of and empirical support for the idea that storytelling evolved to facilitate cooperation. We offer empirical support through: 1) each of the four collected Agta stories concerns social and cooperative behaviour; 2) in a content analysis of 89 stories from seven hunter-gatherer societies, ~70% of stories reflected social behaviour (more than any other category of story); and 3) Agta camps with a greater proportion of skilled storytellers were associated with increased levels of cooperation.

It is important to note that even though we only offered empirical evidence to link the evolution of storytelling with the evolution of cooperation instead of rejecting the other theories, this is a huge empirical advance in relation to previous studies. All the studies mentioned by the reviewer present different hypotheses to explain the evolution of storytelling. However, none of them presented a single case of experimental testing based on data analysis, offered empirical evidence in favour of their own hypothesis, or attempted in any way to empirically reject hypotheses alternative to their own. Although we are familiar with all studies on the list, due to space issues we exemplify our point with the following cases:

Coe et al (2006) "*focused on stories that encourage generosity, cooperation, restraint and sacrifice, because these are directly related to the formation of the social relationships so crucial to our species*". However, they state that "*it may be impossible to test directly whether traditional stories have increased the*

descendant-leaving success of ancestors", and only state that *"the continued existence of the same story for centuries demonstrates that it did not prevent the leaving of descendants"*. Hence, they have neither tested their suggestion of a link between cooperation and storytelling, nor excluded any alternative hypotheses.

Tooby and Cosmides (2001) discussed three explanation for why humans produce fiction (and art in general): 1. because they are *"functional products of adaptations that are designed to produce this engagement"*; 2. because they are a *"by-product of adaptations that evolved to serve functions that have nothing to do with the arts per se"*; or 3. they *"spread by chance during evolution"*. The article does not present experimental tests of any of the hypotheses, but instead presents a verbal argument for why hypothesis 1 is the best. They conclude that *"in sum, we think that art is universal because each human was designed by evolution to be an artist, driving her own mental development according to evolved aesthetic principles"*, but this statement does not disprove alternative explanations.

Mar and Oatley (2008) propose that *"stories simulate or model the social world through abstraction"*, and that *"abstraction of experience found in stories evokes...a simulative experience that allows for the compelling and efficient transmission of social knowledge"*. However, their monograph does not present any test of the hypothesis of 'fiction as simulation'. In addition, no alternative hypothesis is discussed or tested.

Boyd (2009) argues that art is a specifically human adaptation: *"we have evolved to engage in art and in storytelling because of the survival advantages they offer our species. Art prepares minds for open-ended learning and creativity; fiction specifically improves our social cognition and our thinking beyond here and now....[they] alter synaptic strength a little at a time, over many encounters, by exposing us to the supernormally intense patterns of art."* More particularly, he argues that our fondness for storytelling has sharpened social cognition, encouraged cooperation, and fostered creativity. Boyd examines Homer's *Odyssey* and Dr. Seuss's *Horton Hears a Who!*, demonstrating how an evolutionary lens can offer a new understanding and appreciation of specific works. However, there is no attempt to analyse a cross-cultural dataset or present another form of empirical testing to substantiate the claim that there is an evolutionary pattern. In addition, no other hypotheses are tested.

Therefore, we believe that we offer much more empirical evidence in support of one of the hypotheses than the studies cited by the reviewer. We believe that our study goes beyond the current state of the field - a field where, as exemplified by our summaries above, hypotheses are generally proposed without any serious attempt to quantitatively test them through data collection, compilation and analysis.

We nonetheless thank the reviewer for the reference suggestions (which we have cited in our manuscript) and for the comment. We have now made it clear in our manuscript that we offer support for one of the hypothesis, but we do not discard the contribution of other selective pressures. For changes to our manuscript which clarify our position, see below:

Lines 186-189: Alternatively, people might enjoy listening to stories for other reasons, such as a form of 'mental simulation' to learn about their social and physical environment, and be paying for this service (this effect could be independent from the function of storytelling in promoting cooperation).

Lines 206-210: Although narratives are known to serve other adaptive functions, such as disseminating information on survival, foraging and the environment, or simply to entertain and hold the audience's attention, we have shown that storytelling may also have been an important factor in the origins of widespread human cooperation.

In my previous review, I called for more attention to design. This is another area where a review of the literature might be helpful. A case in point is the possible benefits of storytelling to the storyteller and the audience—specifically, the claim that “storytellers may be ‘rewarded’ for their public good by other camp-mates who benefit from the increased cooperation which storytellers may promote, in what may be mutually-beneficial trade-like relationships (although the individual-level benefits to those who cooperate with storytellers remain in need of further study)” (172-174). While I agree that the potential benefits of storytelling need further study—especially quantitative study—there has been some theoretical work on the subject. Since the claim made in the paper is theoretical as well, I don't understand why there is no reference to this previous work--e.g., Pinker 1994 (538-543); Scalise Sugiyama 1996 (411-412), 2005 (188-189); Tooby & Cosmides 2001.

We thank the reviewer for their helpful suggestions and have included these references in an expanded discussion (lines 185-189): ‘...remain in need of further empirical study). Alternatively, people might enjoy listening to stories for other reasons, such as a form of 'mental simulation' to learn about their social and physical environment, and be paying for this service (this effect could be independent from the function of storytelling in promoting cooperation).’

We would like to emphasise that we are aware of a long tradition of theoretical studies and reviews on the topic of storytelling. Our aim was to provide a solid, data-driven study of the subject based on original data from hunter-gatherers and an analysis of 89 stories from a number of hunter-gatherer populations. We agree this is a path to be further extended, and we believe our study is an important step towards an understanding of the origins of storytelling in humans.

On this point, one of the most suggestive findings of the study is that the effect of age was “strongly significant” (308-309), that “older individuals possessed a greater storytelling reputation” (295-296), and that “many of the Agta rated by others as skilled storytellers were also those who were the most engaging and knowledgeable during our fieldwork” (347-348). These findings are consistent with the hypothesis that storytellers are valued because of the knowledge they possess (i.e., all of the knowledge transmitted via the stories, not just information about social norms). Humans continue to accumulate knowledge across the lifespan: thus, the longer a person has lived, the more knowledge he/she possesses. This may be the basis for the deep respect accorded to elders in indigenous oral cultures: although I don't know of any study that has

quantified this relationship, in the ethnographic literature elders are commonly characterized—both emically and etically--as the “libraries” of their society. On this point, it would be useful to know what the participants understood “skilled storyteller” to mean. A person who can tell a story in an engaging manner? A person who knows a lot of stories? A person who knows the stories in their entirety and can recount them without error? All of the above?

We agree that this is an interesting finding and is consistent with many qualitative ethnographic accounts which value elders as repositories of knowledge. While our paper does not address this finding in detail as it is not central to our main argument, it is nonetheless suggestive, although the reasons underlying this pattern are not well-understood. Perhaps, as the reviewer suggests, it is due to accumulated knowledge over one’s lifetime, but may also be due to older individuals less able to contribute to physical endeavours (foraging, etc.), so specialise in social activities such as storytelling. Alternatively, storytelling may be a complex skill requiring years of practice to master. These are all interesting questions for future research to explore in greater detail.

Unfortunately we do not have detailed data regarding perceptions of ‘good storytellers’. Nominations for skilled fishers reflected both skill (returns per hour) and effort (total calories obtained; lines 351-357), and similar factors are likely to have also influenced storyteller nominations – that is, those who tell stories frequently and tell them well. However, further research is required to understand the specific factors which contribute to becoming a ‘skilled storyteller’. Nonetheless, we want to make it clear that, for the purposes of our analyses, it was enough for us to know who was considered a ‘skilled storyteller’ by the Agta themselves, and that we managed to show correlations between number of skilled storytellers and levels of cooperation in camps.

Finally, da Silva & Tehrani (2016) trace the origin of some Indo-European tales back about 6000 years. While this is an exciting study, 6000 years is not a “deep evolutionary history” (63), nor is it sufficient time for a complex cognitive adaptation to evolve. I support the claim that storytelling is ancient, but would recommend citing more compelling evidence of its antiquity.

We agree that 6,000 years is not ‘deep evolutionary history’, and did not intend to imply that a complex adaptation such as storytelling could evolve in this time-span. We have therefore altered this section to read (lines 61-64): ‘Storytelling is a human universal which occurs spontaneously in childhood, while cross-cultural phylogenetic analyses have shown that folk stories may be highly conserved. The universal presence and antiquity of storytelling indicates that it may be an important human adaptation’.

Methods

I completely agree that “it is difficult to obtain an unbiased sample of stories told by hunter-gatherers as different ethnographers tend to focus on different story contents” (451-453). Given that this is the case, then, it is unclear what—if anything--frequency measures of content can tell us about the relative importance of the different types of information contained in collections of forager oral tradition. Moreover, it seems incautious to conclude that, because one type of information occurs more frequently than another, the former

references a selection pressure while the latter does not. This appears to be what the paper is claiming, as evinced by the statement that, "as social content was present in ~70% of stories, approximately double that of the second-highest content (natural phenomena; 36%), it is unlikely that any small changes in categorization would greatly influence our conclusion that stories in hunter-gatherer societies appear to function to coordinate social behavior" (495-499). This seems tantamount to arguing that, because hunter-gatherer women spend more time gathering than giving birth, gathering exerted selection pressure on our female ancestors but parturition did not. To put it another way, because virtually all stories feature characters (i.e., agents), narrative processing involves making inferences regarding the characters' mental states (e.g., beliefs, desires, feelings); thus, a frequency-based approach might lead one to argue, à la Zunshine (2006) and Mar & Oatley (2008), that storytelling organizes theory of mind rather than cooperation.

Even though there are likely to be systematic biases in the stories collected by ethnographers, the finding that most stories contain social content is at the very least consistent with our hypothesis (given that many ethnographers tend to focus on cosmological and religious content it is plausible that our analysis may even underestimate the social content of stories told by foragers). As noted by another reviewer, if few of the stories told by hunter-gatherers concerned social behaviour, then the effect of storytelling on cooperation was likely to be limited. Again, we did not mean to imply that stories do not fulfil other adaptive functions beyond coordinating social behaviour (see our responses above). Around one-third of all stories contain information about the natural environment or resource use, suggesting that stories are also important to impart these kinds of information too. We have therefore updated the sentence the reviewer noted to now read (lines 505-509): 'as social content was present in ~70% of stories, approximately double that of the second-highest content (natural phenomena; 36%), it is unlikely that any small changes in categorization would greatly influence our conclusion that one of the functions of stories in hunter-gatherer societies may be to coordinate social behaviour'.

The analogy the reviewer provides to dismiss our content analysis is also unfair as foraging and parturition are different domains, so a comparison of time-budgets to infer selective pressures for these activities would be inappropriate. Our analysis focuses within a single domain (storytelling), which may give an insight into its adaptive function as individuals can choose to tell stories of one type or another (say, between social or cosmological content). Perhaps a more appropriate analogy for our analysis would be that women from forager societies tend to gather more than they hunt (and vice versa for men), and that these differences reflect different selective pressures, which undoubtedly they do.

We also agree with the reviewer that one of the functions of stories may be to promote perspective-taking, which is an integral mechanism to coordinate group behaviour. In our manuscript we stated that 'by introducing individuals to situations beyond their everyday experience, narratives may also increase empathy and perspective-taking towards others, including strangers, potentially facilitating camp-level coordination and cooperation' (lines 199-202). However, many of these stories do not just detail others' mental states, but have a moral dimension 'which prescribed and coordinated behaviour during interactions with

others' (line 494). This leads us to conclude that one of the functions of stories is to broadcast social norms and represent punishment of norm-breakers.

I am also curious as to how the story content analysis was conducted—especially, how information about animal behavior was operationalized. In forager oral tradition, much information about animal behavior is encoded in origin myths and etiological tales. For example, in a Koyukon story about the origin of the potlatch and the games associated with it, the animal-people have a race. The wolverine is in the lead, but then “each time he saw a little spruce tree he went to rub his butt on it He would have won but he never passed a spruce tree (without stopping to rub his butt on it) So I don't know who won, maybe it was the wolf, it's said that the wolf also travels very fast” (Atlla 1983:55). These lines describe the wolverine's habit of marking its territory using anal scent glands, and also note that both wolverines and wolves can travel rapidly through snow (the story is set in winter). Due to the anthropomorphic aspects of the characters, a naïve coder might miss this information, or mistakenly assume it was fanciful. So my concern is that the content analysis may underestimate the frequency with which information about animal behavior/traits occurs in forager oral tradition. Information about natural disasters, environmental hazards, subsistence stress, and wayfinding is similarly encoded and, consequently, may also be underrepresented in the content analysis.

Stories in the content analysis were coded to be as inclusive as possible, so any mention of animals which may impart knowledge to others about its behaviour – such as the Koyukon story provided by the reviewer – would be coded under the category 'natural phenomena'. We do of course grant that the interpretation of stories is somewhat subjective, and as we state in our manuscript (lines 503-505): 'any analysis of this kind will of course be subject to bias to some extent, such that others may categorise stories differently or their 'emic' interpretation in the specific society may be different'. However, these problems of interpretation concern any analysis of complex qualitative information and 'as social content was present in ~70% of stories, approximately double that of the second-highest content (natural phenomena; 36%), it is unlikely that any small changes in categorization would greatly influence our conclusion that one of the functions of stories in hunter-gatherer societies may be to coordinate social behaviour' (lines 505-509). The reviewer's criticism also cuts both ways, as some more subtle aspects of social information may have gone unnoticed, such that social content may also be underrepresented in our analysis.

As noted above, the finding that “storytelling skill is associated with increased fitness” is exciting, but it doesn't speak to precisely what is valued in the skilled storyteller. Is it his/her ability to entertain others, promote cooperation, provide useful information, or something else? I find it impossible to conclude from these results that storytelling is an adaptation for cooperation per se.

As this is the same criticism levelled towards the finding as to why storytellers are preferred camp-mates, for our response we refer the reviewer to the paragraphs above, beginning 'we believe the reviewer is asking: were storytellers preferred camp-mates – or achieved higher fitness – because they promoted cooperation, or for other reasons, such as transmitting other fitness-relevant information?'.

Finally, to demonstrate the presence of design, a phenomenon must be shown to be species-typical. While a survey of 7 cultures is better than a survey of 1 (and a survey of 89 stories is better than a survey of 4), it is still inadequate to this task (and there is still the problem of 5 cultures in the sample being from the same geographic region). I would recommend using another, less labor-intensive approach, although I do not have any good suggestions at the moment.

We thank the reviewer for recognising the advantages of our cross-cultural, quantitative, hypothesis-driven, data-based, comparative story analysis of 7 populations and 89 stories derived from different geographic regions (including Africa, Southeast Asia and India). In the absence of any other obviously more suitable method (as recognized by the reviewer), we believe our approach is a significant advance in understanding the evolution of human storytelling. While we completely agree that more data is necessary to test any theory regarding the function of narratives in greater detail, we do believe that our data are adequate as a preliminary answer to these questions, especially as no previous studies have quantified story content across different forager societies as we have here.

References

- Attla, C. 1983. As my grandfather told it: Traditional stories from the Koyukuk told by Catherine Attla. Yukon-Koyukuk School District and Alaska Native Language Center.
- Boyd, B. (2009). On the origin of stories. Harvard University Press.
- Coe, K., Aiken, N. E., & Palmer, C. T. (2006, March). Once upon a time: Ancestors and the evolutionary significance of stories. In *Anthropological Forum* (Vol. 16, No. 1, pp. 21-40). Routledge.
- Hagen, E. H., & Bryant, G. A. (2003). Music and dance as a coalition signaling system. *Human nature*, 14(1), 21-51.
- Mar, R. A., & Oatley, K. (2008). The function of fiction is the abstraction and simulation of social experience. *Perspectives on psychological science*, 3(3), 173-192.
- Miller, G. (2000). The mating mind: How sexual selection shaped the evolution of human nature.
- Pinker, S. (1994). How the mind works.
- Scalise Sugiyama, M. (1996). On the origins of narrative. *Human Nature*, 7(4), 403-425.
- Scalise Sugiyama, M. (2017). Narrative. *Encyclopedia of Evolutionary Psychological Science*. Springer.
https://link.springer.com/referenceworkentry/10.1007/978-3-319-16999-6_3316-1
- Tooby, J., & Cosmides, L. (2001). Does beauty build adapted minds? Toward an evolutionary theory of aesthetics, fiction, and the arts. *SubStance*, 30(1), 6-27.
- Zunshine, L. (2006). Why we read fiction: Theory of mind and the novel. Ohio State University Press.

REVIEWERS' COMMENTS:

Reviewer #1 (Remarks to the Author):

I am satisfied with the way the authors addressed my latest points.

As Reviewer 3 withdrew, I have also been asked to comment on authors responses to Reviewer 3's extensive set of requests. Although I am far from being as knowledgeable as Reviewer 3 on the theory of story-telling and cultural evolution, my opinion is that the article is publishable in the current format, with no need to address Reviewer 3's points more than the authors have already done. I am under the impression that Reviewer 3 was after a rather demanding test of the theory that "storytelling is an adaptation that evolved to promote cooperation". Reviewer 3 seemed to imply that for this theory (or any theory) to be proven, (a) causality had to be ascertained, and (b) the unique function of storytelling had to be to increase cooperation. I believe that both (a) and (b) are too demanding claims, and would actually make it virtually impossible to corroborate any theory. This is in particular the case if a researcher had to demonstrate that every other possible side-effect of a variable causally affecting another variable are null (as the list of possible side-effects is potentially infinite). I have already commented that even (a) is impossible to prove within the current dataset, as a more stringent prove of causality would have required longitudinal data or some other type of experimental-induced manipulation of story-telling (such as priming). This does not detract from the importance of this study, provided that one is happy to accept that a test to falsify a theory has been provided, and that the evidence gathered does not reject the theory. I think the authors do exactly that, and this is sound epistemologically. Their pointing out that their evidence is correlational and that they obviously cannot rule out that other alternative mechanisms are at play, helps the reader to better appreciate the strength of such a test, and the obvious limitations of the study. For these considerations, I recommend the publication of the current version of this article.